# Recent Changes to the Hydrological Cycle of an Arctic basin at the Tundra-Taiga Transition

Sebastian A. Krogh and John W. Pomeroy

Centre for Hydrology, University of Saskatchewan, 121 Research Dr., Saskatoon, SK S7N 1K2, Canada

*Correspondence to:* Sebastian Krogh (seba.krogh@usask.ca)

**Abstract.**

The impact of transient changes in climate and vegetation on the hydrology of small Arctic headwater basins has not been investigated, particularly in the tundra-taiga transition region. This study uses weather and land cover observations and a cold regions hydrological model to investigate historical changes in modelled hydrological processes driving the streamflow response of a small Arctic basin at the treeline. The physical processes found in this environment and explicit changes in vegetation extent and density were simulated and validated against observations of streamflow discharge, snow water equivalent and active layer thickness. Mean air temperature and all-wave irradiance have increased by 3.7°C and 8.4 W m$^{-2}$, respectively, while precipitation has decreased 48 mm (10%) since 1960. Two modelling scenarios were created to separate the effects of changing climate and vegetation on hydrological processes. Results show that over 1960-2016 most hydrological changes were driven by climate changes, such as decreasing snowfall, evapotranspiration, deepening active layer thickness, earlier snowcover depletion, and diminishing annual sublimation and soil moisture. However, changing vegetation has a significant impact on decreasing blowing snow redistribution and sublimation, counteracting the impact of decreasing precipitation on streamflow, demonstrating the importance of including transient changes in vegetation on long-term hydrological studies. Streamflow dropped by 38 mm as a response to the 48 mm decrease in precipitation, suggesting a small degree of hydrological resiliency. These results represent the first detailed estimate of hydrological changes occurring in small Arctic basins, and can be used as a reference to inform other studies of Arctic climate change impacts.

## 1   Introduction

Rapid warming in the Arctic (Hansen et al., 2010; Przybylak *et al.*, 2010; Wanishsakpong *et al.*, 2016) has produced significant environmental changes (Hinzman et al., 2005), such as decreasing snowcover duration (Brown et al., 2010) and permafrost thaw (Liljedahl et al., 2016). A reduced snowcover period can result in smaller, slower snowmelt, larger evapotranspiration losses and reduced sublimation losses from cold regions headwater basins (Pomeroy et al., 2015; Rasouli et al., 2015). Permafrost thaw can impact regional and local hydrology by increasing surface and subsurface connectivity, and baseflow (Connon et al., 2014; Liljedahl et al., 2016; Walvoord and Kurylyk, 2016). Increases in vegetation cover and density have been observed and are especially pronounced near the tundra-taiga ecozone transition (Lantz et al., 2013; Myers-Smith et al., 2011; Sturm et al., 2001; Tape et al.,

2006; Xu et al., 2013); however, the impact on the hydrology of these transition Arctic basins is poorly understood. These environmental changes will likely continue in the future, representing challenges for water resources managers and engineers throughout the Arctic.

Precipitation trends over the Arctic are highly uncertain due to a sparse monitoring network (Serreze et al., 2003) and difficulties in measuring snowfall in windswept environments (Goodison et al., 1998; Pan et al., 2016). Nevertheless, positive and negative trends have been calculated for the largest Arctic river basins (Walsh, 2005, table 6.12) and throughout the Arctic (Whitfield et al., 2004). Over northern Canada, an overall increasing trend in annual precipitation has been observed (DeBeer et al., 2016; Vincent et al., 2015); however, there is great spatial variability and uncertainty due to the low-density observational network (Milewska and Hogg, 2001). Mean annual temperatures in northwestern Canada have increased more than anywhere else in Canada by roughly 3 – 3.5 °C between 1948 and 2012 (Vincent et al., 2015); moreover, mean winter temperatures show the largest increase of up to 6.5 °C (DeBeer et al., 2016).

Arctic vegetation has changed in response to warmer temperatures (Hinzman et al., 2005; Martin et al., 2017; Myers-Smith and Hik, 2017). The tundra-taiga treeline in Alaska, U.S.A., has advanced between 80 to 100 m in the last 200 years (Suarez et al., 1999). Payette & Filion (1985) studied white spruce (picea glauca) expansion into northern Quebec, Canada, and found that the treeline has not changed substantially over the past centuries; however, below the treeline, its density has increased. On the other hand, both shrub coverage and density have increased in the Arctic. Lantz *et al.* (2013) reported that between 1972 and 2004, shrub density and cover have increased substantially in the upland tundra east of the Mackenzie River Delta of northwestern Canada. Similar results were found by Tape et al. (2006) in northern Alaska and pan-Arctic. Overall, these previous studies observed that the Arctic treeline has not undergone a substantial change over the last century, but shrub expansion is ubiquitous near the Arctic treeline in North America. Wildfires can rapidly modify vegetation cover and are important to nutrient cycling, biodiversity, and control of pathogens and pests (Bond and Keeley, 2005). Warmer temperatures and longer dry seasons are increasing vulnerability to wildfire (Romero-Lankao et al., 2014), resulting in increased frequency and duration of wildfires since the mid-1980s (Westerling et al., 2006; Williamson et al., 2009). Changes in vegetation are important, as they have been shown to control snow redistribution (Ellis et al., 2013; Essery and Pomeroy, 2004; Ménard et al., 2014; Pomeroy and Brun, 1990) and energy fluxes (Ménard et al., 2012; Pomeroy et al., 2006; Sturm et al., 2000).

Many studies have looked at observed changes in large northward flowing river basins. There is an increase in annual discharge from large river basins to the Arctic Ocean (McClelland et al., 2006; Overeem and Syvitski, 2010; Peterson et al., 2002; Rood et al., 2017; Yang et al., 2002), a decrease in river ice thickness (Peterson et al., 2002) and an earlier river/lake ice break-up dates (Janowicz, 2010; Prowse et al., 2011). However, most of these large river basins have their headwaters and primary zones of runoff generation well below the Arctic Circle; and therefore are not necessarily representative of changes in the Arctic hydrological cycle. As limited observations are available in the Arctic, model outputs have also been used to investigate change. Increasing trends were found in simulated monthly evapotranspiration and streamflow for the Mackenzie River Basin, Canada (Yip et al., 2012) and in

simulated Arctic soil temperature and active layer thickness (Oelke et al., 2004), and decreasing trends were found in simulated Arctic snow accumulation and snowcover duration (Liston and Hiemstra, 2011). López-Moreno *et al.* (2016) analyzed simulated historical trends in the snow processes of a small basin above the Arctic treeline in Svalbard, using a physically based cold regions hydrological model that accounted for blowing snow redistribution and energy balance snowmelt. They found that simulated snow accumulation, snow-covered season and days with snowfall decreased significantly, driven by a significant increase in air temperature. No study has looked at changes in Arctic hydrological processes from headwater basins that originate near the Arctic treeline, nor has the relationship between changes in hydrological processes due to climate and vegetation change been investigated.

Using hydrological models to simulate the hydrological cycle can compensate for an inability to observe it due to ungauged basins (Pomeroy et al., 2013b) and decline in the coverage of Arctic monitoring networks (Laudon et al., 2017). Previous studies acknowledged the need for robust cold regions hydrological models to simulate Arctic hydrology (Quinton and Carey, 2008; Woo et al., 2008), particularly due to the complex interaction between subsurface and surface mass and energy fluxes (Kane et al., 1991; Krogh et al., 2017; Zhang et al., 2000). Physical processes that must be accounted for include: snow accumulation and melt (Marsh *et al.*, 2010), snow interception and sublimation from forest canopies (Hedstrom and Pomeroy, 1998; Pomeroy et al., 1998; Schmidt and Gluns, 1991), blowing snow sublimation and redistribution (Pomeroy et al., 1997; Schmidt, 1982), evapotranspiration (Wessel and Rouse, 1994), infiltration into frozen and unfrozen soils (Gray et al., 2001; Kane, 1980; Kane and Stein, 1983), water flow through snowpack (Colbeck, 1972; Marsh and Woo, 1984a, 1984b), ground freeze and thaw (Juminikis, 1977), surface and subsurface flow (Quinton and Gray, 2001; Quinton and Marsh, 1999), groundwater (Cederstrom et al., 1953) and streamflow routing (Woo and Sauriol, 1980). The Cold Regions Hydrological Model (CRHM) platform was used to create the Arctic Hydrology Model (AHM) configuration (CRHM-AHM) by Krogh *et al.* (2017). This spatially distributed and physically based model includes the key hydrological processes found at the Arctic treeline, such as blowing snow, snow and rain interception, sublimation, snowmelt, flow through snow, infiltration to frozen and unfrozen soils, evapotranspiration, runoff as overland flow and subsurface flow through organic terrain, frozen ground dynamics including active layer thaw, groundwater flow and streamflow routing. CRHM-AHM was shown to properly represent the winter and summer hydrology of this environment with minimal calibration of some uncertain routing and storage parameters (Krogh et al., 2017). A great advantage of this model is its flexibility and potential to be adapted for simulation of other Arctic basins.

The aim of this study is to understand, diagnose and quantify the long-term hydrological changes of a small Arctic treeline basin, including transient changes over a multidecadal period, using the CRHM-AHM model. The study addresses the following research questions: what hydrological changes are caused by individual transient changes in climate and vegetation? What are the coupled hydrological impacts of changes in climate and vegetation; does transient vegetation change enhance or dampen climate change? Does Arctic hydrology show resiliency to the impacts of climate change? To address these questions, the study compares three observation-driven hydrological modelling scenarios: (1) observed climate change and constant vegetation cover, (2) observed transient changes in vegetation with climate held constant and (3) observed changes in climate and vegetation.

## 2    Study site

Havikpak Creek (HPC) with an area of 16.4 km$^2$ is located east of Inuvik, Northwest Territories, Canada (Figure 1), near the tundra-taiga transition. HPC is in the continuous permafrost region, with an elevation rising from 60 m.a.s.l. in the southwest to 240 m.a.s.l. in the northeast. This basin was selected as it has a history of process-based hydrological studies, which provides a good understanding of dominating hydrological processes, has long-term meteorological records and has been part of important international initiatives, such as the Mackenzie GEWEX study (MAGS). HPC is also within the domain of the NASA Arctic-Boreal Vulnerability Experiment (ABoVE; https://above.nasa.gov/), which aims to better understanding the vulnerability and resiliency of Arctic boreal ecosystems, and therefore, its great relevance.

Estimates of mean annual temperature and precipitation between 1981 and 2010 at Inuvik, using observations at the Meteorological Service of Canada weather station (Climate ID: 2202570) by Environment and Climate Change Canada (ECCC), are -8.2 °C and 240.6 mm, respectively (http://climate.weather.gc.ca/climate_normals/index_e.html). However, Krogh *et al.* (2017) showed that the corrected mean annual precipitation between 1980 and 2009 at Inuvik, based on the Adjusted and Homogenized Canadian Climate Dataset (AHCCD; Mekis and Vincent, 2011) and additional local weather stations, is 327 mm. Differences between precipitation estimates published by ECCC and Krogh *et al.* (2017) are due to Krogh's use of the AHCCD dataset with its corrections of snowfall wind undercatch and trace events. Such large adjustments to corrected precipitation are not uncommon at high latitudes in Canada and can influence trend detection (Pomeroy and Goodison, 1997). Using corrected data, 59% of the mean annual precipitation is snowfall; however, peak monthly precipitation occurs as rainfall in August (~45 mm; Krogh *et al.*, 2017). Snow accumulation typically starts in mid-September, with peak accumulation at the end of April or beginning of May, and snowmelt lasts until early- to mid-June (Krogh et al., 2017). The streamflow regime of Havikpak Creek is measured by the Water Survey of Canada and is characterized by a rapid increase due to snowmelt in May and June, during which the annual peak streamflow occurs (1-4 m$^3$ s$^{-1}$), followed by decreasing streamflow interrupted by sporadic summer peaks due to intense rainfall (Krogh et al., 2017). No streamflow was observed during winter.

In 1992, HPC was predominantly covered by Black Spruce (*Picea mariana*) forest (50.0%) followed by Alder shrubs (31.7%), short grass, moss and lichen tundra (11.6%) and open water (6.7%) (Krogh et al., 2017). However, as shrubs colonize the tundra (Lantz et al., 2013) these percentages have changed. No changes in forest cover have been reported, though investigations into this are understood to be underway. A slight "greening" of the region has been detected through NDVI analysis of Landsat satellite imagery, but not attributed to specific vegetation changes (Ju and Masek, 2016). Soils in HPC are characterized by a top layer (roughly 10 cm) of decomposed and highly porous organic matter (upper peat), followed by a highly decomposed and denser organic layer underneath (lower peat), estimated between 20 to 50 cm thick on top of a mineral soil layer (Krogh et al., 2017). No soil changes have been reported. For a detailed description regarding HPC climate, landcover, soils, weather and hydrometric stations, the reader is referred to Krogh *et al.* (2017).

## 3    Data

Reconstructed weather time series used in this study are based on a combination of observations, adjusted and homogenized time series from the AHCCD dataset (station ID: 2202578; Mekis and Vincent, 2011), ERA-40 (Uppala et al., 2005) and ERA-Interim (Dee et al., 2011) reanalysis over the period 1960-2016. Reanalysis data has
5    been used in the past to complement meteorological observations for hydrological studies (e.g. Krogh et al., 2015). Six hourly-timestep variables were used to drive CRHM-AHM (see section 4.1): precipitation, air temperature, relative humidity, wind speed, and incoming short- and long-wave radiation (Figure 2). Data used for model validation consist of observed daily streamflow (section 5.2). A reconstructed vegetation cover map, topographic information and a site visit informed the spatial model configuration.

### 3.1    Temperature

Daily minimum, maximum and mean temperatures from the AHCCD dataset are available from 1957 to 2014. Hourly temperature is available from Inuvik Airport from 1980 to 2016 and from Inuvik Climate from 2003 to 2016. To generate a uniform time series of hourly temperature, the following steps were followed: (1) minimum and maximum from AHCCD dataset (1957-2014) were used to generate hourly temperature by fitting a sinusoidal
function, as presented by Chow and Levermore (2007; Equation 6); (2) hourly temperature measured by the Inuvik Airport station (1980-2016) was used to correct hourly temperature from the AHCCD dataset (1960-1980) through a linear regression model ($R^2 = 0.97$); and (3) Inuvik Airport hourly data was used for the period 1980-2016.

### 3.2    Precipitation

Daily precipitation from the AHCCD dataset for the period 1960-2006 is available; however, after 1994, several
gaps were found. Precipitation measurements from the AHCCD at Inuvik were all made by observers and are considered reliable. After 1994, automatic systems were sometimes used to improve the corrections from snow ruler measurements (Mekis and Vincent, 2011). For measurements from 1994 to 2007, a combination of AHCCD and the local ECCC automatic weather stations: Inuvik Climate, Inuvik Upper Air and Inuvik Airport (Figure 1), was used. From 2007 onward, the Inuvik Climate station (automatic) was the only station recording precipitation. The
automatic station snowfall data was corrected for wind undercatch using the expression presented by (Smith, 2008) for the Alter-shielded Geonor solid precipitation gauge. A specific snowfall correction had to be applied between October and March for the water years 2010 to 2012, as winter precipitation from the Inuvik Climate precipitation gauge was not found to be credible. Observed snow accumulation (snow water equivalent, SWE) in sheltered sites and observed streamflow suggest that snowfall measured during these years was grossly underestimated. The ratio
between measured end-of-the-winter SWE (April 1st) and cumulative snowfall in 2011, 2012 and 2013 was 2.6, 1.7 and 2.5, respectively (after wind undercatch corrections); the ratio associated with the other years with both SWE and streamflow data (2003 to 2015) show values around 1. A solution to this problem was proposed and implemented by Pomeroy *et al.* (1997) at a nearby location (Trail Valley Creek) and consists of estimating 'true' winter snowfall from late season snow surveys in a small glade within a forest. Pomeroy *et al.* (1997) argued that the
wind and sun sheltered, and cold conditions of the site ensured that the snow on the ground in the glade was not

redistributed, sublimated or melted, and was therefore equal to the cumulative snowfall. SWE measurements used in this study have the same conditions as those found by Pomeroy *et al.* (1997) (i.e. sheltered site with mild winds and cold environment), and therefore, their approach was used to estimate 'true' snowfall.

To disaggregate daily into hourly precipitation, the same procedure used in Krogh *et al.* (2017) was followed. This employs the microcanonical cascade model presented by Güntner et al. (2001). This disaggregation technique assumes that the probability distribution function of the weights factors, defined as the ratio between a lower and upper disaggregation level (e.g. 12 hr and 24 hr), from the different disaggregation levels (e.g. 3, 6, 12 and 24 hr) is constant, and it was obtained aggregating hourly precipitation records. The reader is referred to Güntner et al. (2001) and Krogh *et al.* (2017) for further details of this methodology and the particular application to Inuvik precipitation dataset, respectively.

### 3.3    Relative Humidity

Relative humidity was calculated using six-hourly air temperature and dew point temperature from ERA-40 for the period 1960-2002, using the expression from Lawrence (2005). A linear interpolation was then used to calculate hourly values. ERA-40 values from 1960-1980 were corrected using a linear relationship for the period 1980-2002 between hourly ERA-40 and measured relative humidity at Inuvik Airport ($R^2$ = 0.7). Finally, hourly corrected values from ERA-40 were used from 1960-1980 and observed values from 1980 to 2016. Relative humidity was not permitted to exceed 100% in this estimation.

### 3.4    Wind speed

Hourly 10-metre height wind speed from the AHCCD dataset and Inuvik Airport station for the period 1960-2006 and 2006-2016 were used, respectively.

### 3.5    Short- and long-wave irradiance

Short- and long-wave irradiance were not measured and so were obtained from the ERA-40 (Uppala *et al.*, 2005; 1960-2002) and ERA-Interim (Dee *et al.,* 2011; 1979-2016) atmospheric model reanalyses at three-hr time steps. A linear interpolation was used to obtain hourly values for each dataset. The ERA-Interim is a more advanced reanalysis, and has shown small biases in Arctic environments (Lindsay et al., 2014), so it was used as "true" incoming radiation and ERA-40 outputs were corrected to match the ERA-Interim. The overlapping period between ERA-40 and ERA-Interim is 1979-2002 (23 years); this period was used to bias-correct ERA-40 over 1960-1979 using the quantile mapping technique. Quantile mapping is a statistical approach used in hydrometeorological studies to bias correct weather variables times from atmospheric models against measurements (e.g. Boé *et al.*, 2007); it corrects each quantile by matching the empirical cumulative distribution functions. The irradiance time series created uses the bias-corrected ERA-40 for 1960-1979 and ERA-I for 1979-2016.

### 3.6    Streamflow

Daily streamflow discharge at HPC was observed and estimated at the hydrometric station (ID: 10LC017) by the ECCC Water Survey of Canada (WSC). This station is downstream from the Havikpak Creek crossing with the Dempster Highway and its drainage area defines the basin for modelling purposes. Discharge estimates for this station start in 1995 and are available to 2015; however, the year 2005 is not available. Measuring small stream discharge in the Arctic is challenging and problems or uncertainties associated with the estimates are acknowledged in the metadata provided by the ECCC through the Environment Canada Data Explorer. The main issues in the hydrometric record are due to the presence of ice and snow in the cross section during snowmelt including peak streamflows, as ice and snow cause substantial variability in rating curves and make streamflow and water stage measurements quite difficult.

### 3.7    Vegetation cover and shrub density

The vegetation cover map and shrub density used in this study are based on the map and values presented by Krogh *et al.* (2017) from 1992, and the changing shrub cover and density rates presented by Lantz *et al.* (2013) for a larger region that includes Havikpak Creek. Lantz *et al.* (2013) showed that between 1972 and 2004 (32 years) shrub cover increased by 15% (±3.6) and shrub density increased by 68% (±24.1), in average. These average rates were recalculated to an annual basis, resulting in rates of 0.47 % yr$^{-1}$ and 2.13 % yr$^{-1}$ for shrub cover and density increases, respectively. To reconstruct times series of vegetation cover and shrub density, the average rates presented by Lantz *et al.* (2013) were used to linearly extrapolate forwards and backwards from the values used in Krogh *et al.* (2017), creating a time series of vegetation cover and shrub density from 1960 to 2016. As shrubs colonize the tundra, any increase in shrub cover is compensated by a decrease in the tundra cover, maintaining a constant drainage area. It is unclear when shrubs expansion in the Arctic began (Tape et al., 2006), mostly because satellite images started to be available in the 70's, limiting out understanding of vegetation changes to the 70's onward.

The HPC forest was held constant in this study, as there are no published studies quantifying forest cover or density change in the region. However, we acknowledge that there are ongoing investigations about changes in forest structure in the region. Greening of the NDVI is not directly attributable to forest change and could be due documented shrubification. There are no recorded wildfires in Havikpak Creek during the study period as it is close to the airport and so fire suppression by local authorities is very effective.

## 4    Methodology

### 4.1    Hydrological modelling

The Cold Regions Hydrological Modelling platform (CRHM) is a process-based and spatially distributed hydrological modelling-system with a flexible modular structure that allows the selection of different hydrological processes from an extensive library to create a customized hydrological model. Most of the modules available in the CRHM have a strong physical basis, with particular emphasis on, but not restricted to, cold region processes. The

CRHM Arctic Hydrology Model configuration (CRHM-AHM) developed and verified by Krogh et al. (2017) includes the following hydrological processes: forest canopy interception, sublimation and evaporation, snow-melt and snow accumulation, evapotranspiration, blowing snow redistribution and sublimation, ground freeze and thaw, water flow through snowpack and organic terrain, infiltration into frozen and unfrozen soils, soil moisture storage and flow, surface water flow and streamflow routing. The model was run over October 1960 to October 2016 on an hourly basis. A four-year spin-up period was used by repeating the years 1960-1963.

CRHM uses Hydrological Response Units (HRUs; Flügel, 1995) as the spatial unit of discretization for application of the continuity equation to compute mass and energy fluxes. In the CRHM-AHM, HPC basin was discretized into 11 HRUs initially classified by land cover: tundra, sparse shrubs, close shrubs, taiga, forest, wetland, and open water. To include the different near-surface wind regimes observed by Pomeroy and Marsh (1997) over the basin, the tundra and sparse shrubs HRUs were each split into an upper and lower HRU to reflect stronger wind regimes in the hilly, higher elevation, upper basin. To simulate the long lasting snow drifts found in steep gullies and around small lakes, a Gully/Drift HRU was created following the criteria from Pomeroy and Marsh (1997). The physiographic characteristics of the HRUs used in the CRHM-AHM applied in HPC are as in Krogh et al. (2017, Table 2).

The parameterization of the CRHM-AHM followed the Deduction-Induction-Abduction approach (DIA; Pomeroy et al., 2013a) by first using field information (e.g. slope and vegetation cover) parameters from previous studies in Havikpak Creek and other research basins with similar hydrological regimes and physical processes to set parameters, and then calibrating against streamflow a few subsurface and surface hydraulic and storage parameters for which there was poor understanding. The CRHM-AHM represents the snow, permafrost and streamflow regimes of Havikpak Creek well when compared to observations (Krogh *et al.*, 2017).

### 4.2    Modelling scenarios

Three modelling scenarios representing only historical climate change (ΔC), only historical vegetation change (ΔV) and both historical climate and vegetation change (ΔCV) were developed to examine the hydrological impacts of changes in HPC since 1960 and are described below.

#### 4.2.1    Model Scenario 1 (ΔC): changing climate and constant vegetation

This scenario uses the reconstructed climate time series presented in section 3 for the period 1960-2016 with a constant vegetation cover and density representative of the year 1988, which is the average vegetation cover of the modelling period.

#### 4.2.2    Model Scenario 2 (ΔV): constant climate and changing vegetation

This scenario uses a "normal" water year in terms of precipitation and temperature to generate the stationary climate. The mean annual (October to September) precipitation and temperature for the period October/1960 to October/2015 is 332 mm and -8.2 °C, respectively. To select a "normal" water year, the residual between mean

annual precipitation and air temperature for the entire period (1960-2016) was calculated to select water year with the minimum combined residual. This was the water year 1962-1963 as the mean annual precipitation and temperature were 327 mm and -8.0 °C, respectively. Seasonal representability was also investigated by looking at the standard deviation of the absolute difference between mean monthly values and the 1962-1963 water year monthly values, resulting in a 10 mm and 1.1 °C for precipitation and temperature, respectively, suggesting that 1962-1963 is a good representation of the monthly variations. Given the importance of snowmelt to streamflow in the Arctic, winter precipitation (October to April) was compared; for 1962-1963 it was 194 mm, and on average over the period it was 166 mm, suggesting that this "normal" year is somewhat snowier than average.

This scenario includes transient changes in vegetation using the vegetation cover and density time series as described in section 3.7. The increase in shrub cover was proportionally applied to the Upper and Lower Sparse Shrubs HRUs, whereas the area of the Wetland and Gully/Drift HRUs were kept constant as their delineation does not depend on the shrub covered area, but on wetland and topographic criteria (Krogh et al., 2017). To implement this transient change, the model was run annually and the shrub cover and density parameters were incremented every November 1[st]. Figure 3 presents the change in area for the Sparse Shrubs and Tundra HRUs during the modelling period, and the year 1992, which is the vegetation cover used by Krogh *et al.* (2017).

### 4.2.3 Model Scenario 3 (ΔCV): changing climate and vegetation

This scenario includes changing climate and transient vegetation as presented for the scenarios ΔC and ΔV, and represents the hydrology of Havikpak Creek as realistically as possible.

### 4.2.4 Transferring initial conditions

In ΔV and ΔCV, the CRHM-AHM was run annually to permit the updating of vegetation parameters at the end of the hydrological year; therefore, final conditions from one year needed to be transferred to the next, and updated with the change in the HRU area. To transfer the initial condition of a given state variable "S" (e.g. volumetric soil moisture or snow water equivalent) from the year (t) and HRU1 ($S_1^t$) to the next year and HRU2 ($S_2^{t+1}$), the following relationship can be obtained through mass conservation, assuming that area is transferred from HRU1 (Tundra) to HRU2 (Sparse Shrub):

$$S_2^{t+1} = \frac{A_2^t * S_2^t + (A_1^{t+1} - A_1^t) * S_1^t}{A_2^{t+1}} \qquad \text{Equation 1}$$

$$S_1^{t+1} = S_1^t \qquad \text{Equation 2}$$

Equations 1 and 2 were used to pass on soil moisture, soil recharge and snow water equivalent state variables from year to year as HRU areas changed.

### 4.3 Trend and change point analysis

The non-parametric Mann-Kendall test (Kendall, 1975; Mann, 1945) was used to perform trend analysis on simulated hydrological variables and observed weather data using a significance threshold of $p \leq 0.05$. The Mann-

Kendall test has been extensively used to analyse linear trends in hydrological datasets (e.g. Burn and Hag Elnur, 2002; Hamed, 2008; Yip *et al.*, 2012), proving better results than other methods (Hess et al., 2001). As recommended by Hamed and Rao (1998) time series autocorrelation was removed before performing the Mann-Kendall test to eliminate the detection of false trends. The trend of slopes was calculated using Sen (1968) based on Kendall's rank correlation τ. Variables presented as a percentage of annual precipitation (i.e. rainfall and snowfall ratios) were log transformed (y = log (x / (1-x))) first. Single change point in the time series were detected using the R-Package "changepoint" version 2.2.2 (Killick et al., 2016) based upon (Hinkley, 1970). These two techniques (Mann-Kendall and change point analysis) were used together as they complement each other and can be used to look at changes in different ways. For example, the detection of significant trends using Mann-Kendall depends on the arbitrary significance threshold, whereas the change point analysis assumes that the time series is normally distributed. Although both techniques have their own limitations they are both equally legitimate, resulting in potentially two different results, such a time series with no statistically significant trend but a detectable mean change point.

## 4.4    Teleconnections

To determine the influence of climatological teleconnections on hydrometeorological conditions in HPC, basin-scale mass fluxes were correlated to five climatic indexes representing large scale circulation features over 1960-2016: (1) Arctic oscillation (AO; Thompson and Wallace, 1998), (2) North Atlantic Oscillation (NAO; Hurrell *et al.*, 2001), (3) North Pacific Index (NPI; Trenberth and Hurrell, 1994), (4) Southern Oscillation Index (SAO; Rasmusson *et al.*, 1982) and (5) Pacific Decadal Oscillation (PDO; Mantua and Hare, 2002). These climatic indexes have been used to investigate teleconnections in Arctic and subarctic environments (Bonsal et al., 2006; Déry and Wood, 2004; Serreze et al., 2002). Teleconnections analysis was restricted to ΔCV as this scenario fully represents observed change in HPC.

## 5    Results

### 5.1    Meteorological trends

Figure 4 shows point changes and trends in seasonal and water year (October to September) precipitation for the period October 1960 - October 2016. Seasons were defined based on local hydrology: winter is from October to April when the snowpack forms and redistributes, spring is May when most snowmelt occurs, summer is from June to August and is a season of rainfall, soil thaw and minimal snowmelt, and fall is September when the active layer of the grounds starts to refreeze and precipitation shifts to snowfall. No trends were found for seasonal or annual precipitation, except spring, which had a significant and decreasing trend of -2.7 mm decade$^{-1}$. Conversely to the trend analysis, the change point analysis shows changes at most seasons and annually. Winter, spring and summer precipitation decreases from 187 to 160 mm, 25 to 13 mm, and 146 to 108 mm, respectively; whereas, fall precipitation increases from 16 to 34 mm. Annual precipitation decreases from 369 to 321 mm (48 mm) at the water year 1972 (Table 3). Analysis of the number of days with precipitation above the thresholds 1, 2, 5, 10 and 25 mm

day$^{-1}$ showed a decreasing trend for events greater than 1, 2 and 5 mm day$^{-1}$ with a slope of -3.8, -1.7 and -0.7 days decade$^{-1}$, respectively. There are no changes in measurements methods associated with these changes.

Figure 5 shows seasonal and annual changes points and trends for minimum, maximum and mean daily air temperature. Increasing trends for mean air temperature were found annually and in every season, with the largest positive trend of 0.9 °C decade$^{-1}$ in winter. Maximum air temperatures increased significantly annually and in summer, at 0.3 °C decade$^{-1}$ in both cases. Winter, spring and fall maximum air temperatures did not show significant trends. Minimum air temperatures increased rapidly annually and in winter, at 1.4 °C decade$^{-1}$ in both cases. Spring, summer and fall minimum annual temperatures did not show significant trends. Change point analysis showed that these trends are reflected by an increase in mean annual temperature during the water year 1992, from -9.1 to -7.1 °C (Table 3). Seasonally the change point analysis shows warming at all seasons but in summer and fall for minimum and mean temperature, respectively. Table 1 presents the changes in temperature for the period 1960-2016 for variables with statistically significant trends. The 8 °C increase in annual and winter minimum temperatures and 3.2 °C increase in annual (5.2 °C winter) mean temperatures over 56 years are remarkable and amongst the highest recorded on Earth.

Table 2 presents the statistically significant trends for the other meteorological forcing variables used by CRHM-AHM at seasonal and annual scales. Mean annual short-wave irradiance has been decreasing by -1.4 W m$^{-2}$ decade$^{-1}$ driven by decreases in spring and summer, whilst mean annual long-wave irradiance has been increasing by 2.9 W m$^{-2}$ decade$^{-1}$ with greater increases in summer and fall than in winter and spring. Mean annual all-wave irradiance (short- and long-wave irradiance) has been increasing by 1.5 W m$^{-2}$ decade$^{-1}$; however, summer all-wave irradiance has been decreasing by -2.9 W m$^2$ decade$^{-1}$. Mean annual wind speed did not change and relative humidity has been increasing by 0.8 % decade$^{-1}$. Table 3 shows the change point analysis for the atmospheric variables forcing CRHM-AHM. Mean annual short- and long-wave irradiance have change points in the water year 1969, from 112 to 104 and 230 to 242 W m$^{-2}$, whereas all-wave irradiance has a change point in 1997, from 344 to 348 W m$^{-2}$. Mean annual relative humidity has a change point in the water 2013, from 69 to 75%. No change point was found for mean annual wind speed. Three wind speed thresholds representing non-blowing snow (2 m s$^{-1}$), light drifting (6 m s$^{-1}$) and strong blizzards (12 m s$^{-1}$) were analyzed. Significant decreases in the hours of events larger than 2 and 6 m s$^{-1}$ were found at -71 and -23 events decade$^{-1}$. The number of hourly events with strong blizzards showed no significant trend.

### 5.2    Updated CRHM-AHM Validation

The 1995 to 2015 Nash-Sutcliffe Efficiency (NSE) and mean bias were found to be 0.40 and 6%, respectively; suggesting that the model's streamflow performance is consistent with that showed by Krogh *et al.* (2017), and changing vegetation dynamic parameterization has a small impact on the short-term model's streamflow performance.

### 5.3 Trends comparison between modelling scenarios

### 5.3.1 Sub-basin scale

Figure 6 presents trends in annual (water year) evapotranspiration and sublimation for various HRUs. Evapotranspiration (ET) refers to the actual wetted surface and canopy intercepted rain evaporation and plant transpiration as calculated by Penman-Monteith (P-M) and Priestley-Taylor (P-T; wetlands and lakes) methods (Krogh et al., 2017), but restricted by not only stomatal conductance in P-M but also by available storage of intercepted rainfall, ponded surface water and soil moisture content and the soil moisture withdrawal curve in CHRM. ET in ΔC and ΔCV has been significantly decreasing by 2 and 5 mm decade[-1] for some HRUs, whereas in ΔV it has been increasing from HRU#3 (Upper Gully/Drift). Evaporation from canopy rainfall interception has been decreasing by up to 2 mm decade[-1] in ΔC and ΔCV for most HRUs, but has no trend in ΔV where only vegetation increases. Soil moisture- restricted and -unrestricted ET from P-M and P-T equations has virtually the same trends, except from Taiga Forest (HRU#5), suggesting that soil moisture content has had little effect in ET. Blowing snow sublimation has a decreasing trend in the Upper and Lower Shrub HRUs for ΔV and ΔCV where vegetation increases, with the largest trend in the upper basin (~-14 mm decade[-1]). Decreasing blowing snow sublimation by 3 mm decade[-1] was found in the Upper Tundra HRU for ΔC and ΔCV. Sublimation from canopy intercepted snow has a decreasing trend for all of the vegetated HRUs in ΔC and ΔCV, with the largest trend in the Forest HRU (roughly 6 mm decade[-1]). Sublimation at the snow surface has a decreasing trend in the Forest (HRU#6) in ΔC and ΔCV (~-1 mm decade[-1]), whereas in ΔV it has an increasing trend in the Upper Gully/Drift (HRU#3) and decreasing in the Upper Shrub HRU. Annual sublimation, defined as the sum of the previous three sublimation terms, has a decreasing trend in ΔV for the Upper and Lower Shrubs and Upper Gully/drift (about 2 to 3 mm decade[-1], respectively). In ΔC and ΔCV it has a decreasing trend in the forested HRUs and Lower Shrub HRU, driven by the decreasing sublimation from canopy interception, which is the dominant sublimation term over the basin (Krogh et al., 2017). Blowing snow redistribution, defined as the divergence between incoming and outgoing blowing snow transport, decreased in the Upper and Lower Gully/Drift HRU for all scenarios, between -20 and -45 mm decade[-1] in the upper basin and -10 and -20 in the lower basin.

Figure 7 present a series of trends related to snowcover and ground freeze/thaw. Maximum SWE for ΔC decreased in some HRUs, with the largest trend in the Lower Sparse Shrub HRU (-17 mm decade[-1]), whereas for ΔCV the largest decreasing trend was found in the Upper Sparse Shrub HRU (-54 mm decade[-1]). Maximum SWE for ΔV showed increasing and decreasing trends in the Sparse Shrub and Gully/Drift HRUs, respectively, with the largest changes found in the upper basin. Note that increasing vegetation cover and density hampered blowing snow transport from Sparse Shrub towards Gully/Drift HRUs (Figure 6). The snowcover depletion date for ΔC and ΔCV advanced in almost all HRUs, around -1 and -3 days decade-1, whereas for ΔV, both advancing and retreating were found in the Upper Sparse Shrub and Upper Gully/Drift, respectively. Snowcover duration for ΔC and ΔCV declined for some HRUs (around -1 and -3 days decade[-1]), whereas for ΔV, both extension and decline was found in the Upper Sparse Shrub and Gully/Drift HRUs (roughly 1 and -1 days decade[-1]). Ground thaw initiation had similar changes as the snowcover depletion timing, which is expected as ground thaw typically starts once the ground is

snow-free and temperatures are above 0°C. Active layer thickness (ALT) for ΔC and ΔCV deepened throughout the basin at between 2 and 5 cm decade[-1], whereas for ΔV it deepened in the Upper Sparse Shrub and Gully/Drift HRUs (<2 cm decade[-1]). Snow ablation rate, here defined as the ratio between the maximum SWE and the number of days between maximum SWE and the depletion of snowcover, decreased for ΔC and ΔCV in some HRUs by between -0.1 and -1.5 mm day[-1] decade[-1], whereas for ΔV it increased in the Sparse Shrub and decreased in the Gully/Drift HRUs.

### 5.3.2    Basin scale

The primary annual mass flux trends from the three modelling scenarios are presented in Figure 8 at the basin scale. No trend was found for annual rainfall depths; however, a decreasing trend of -7.8 mm decade[-1] was found for snowfall depths in ΔC and ΔCV. The rainfall ratio (rainfall divided by total precipitation) exhibited no trend. Annual sublimation losses decreased by -1.3, -0.7 and -1.8 mm decade[-1] in scenarios ΔC, ΔV and ΔCV, respectively. The sublimation trend in ΔC was driven by decreasing sublimation from canopy interception, likely due to decreasing snowfall. Decreasing sublimation in the ΔV scenario was driven by decreasing blowing snow sublimation caused by expanding and densifying tundra shrubs, whereas for the ΔCV scenario, both drove sublimation trends. Annual ET losses decreased by -2.5 mm decade[-1] in ΔCV, in contrast to the trend to increase by 0.06 mm decade[-1] for ΔV, driven by positive trends in all ET components. ET in ΔC showed no trend. Decreasing ET in ΔCV was driven by a decreasing trend in evaporation of rain intercepted in the canopy. To investigate the potential impact of changes in stomata resistance on evapotranspiration, trends in mean annual stomata resistance were also calculated. For both scenarios with changing climate (ΔC and ΔCV), no trend was found; however, for the changing vegetation-only scenario (ΔV) a positive trend of 1.6 s m[-1] decade[-1] was found, which agrees with the small increase in ET found for this scenario. Annual streamflow shows an increasing trend of 0.6 mm decade[-1] only for ΔV, likely due to the increasing snow accumulation at some HRUs (Figure 6) as a result of reduced blowing snow transport.

Table 4 presents the change point analysis for selected annual mass fluxes at the basin scale for the three scenarios. Rainfall shows an increase from 131 to 196 mm in 2013, whereas snowfall decreases from 211 to 169 mm in 1997. Similarly to the trend analysis, sublimation shows a decrease in all the modelling scenarios, from 39 to 28 mm, 37 to 35 mm, and 42 to 36 mm, for ΔC, ΔV and ΔCV, respectively. ET, which showed no significant trend for ΔC, presents a decreasing change point from 160 to 144 for ΔC, driven by the dryer conditions. ET for ΔV shows no change point, despite the small significant trend in ET (0.06 mm decade[-1]). For the combined scenario (ΔCV) ET shows a decrease change point in 1977 from 160 to 144 mm, driven by dryer conditions and the decrease in radiative energy for ET (all-wave irradiance). Streamflow for ΔC has a decreasing change point from 180 to 140 mm in 1973, despite the lack of significant trend (Figure 8). For ΔV, streamflow has a small increase from 133 to 135 mm in 1992, which somewhat counteracts the effect of changing climate (180 to 140 mm), resulting in a smaller change from 178 to 140 mm in 1973 for ΔCV.

### 5.4    Streamflow regime change

The ΔCV scenario most comprehensively represents historical change in climate and vegetation in the Havikpak Creek Basin; therefore, it was used to estimate and diagnose changes in streamflow. Figure 9 presents annual time series of variables associated with annual streamflow and peak streamflow for the water years between 1960 and 2015. These time series are: annual streamflow volume (Figure 9a), annual peak daily streamflow discharge (Figure 9b), date (day of the year, DOY) of peak discharge (Figure 9c), the DOY of the centre of mass (50% of volume passed) of streamflow discharge (Figure 9d) and daily streamflow discharge associated with different exceedance probabilities: 5%, 25%, 50%, 75% and 95% using a Weibull distribution function (Figure 9e). The Weibull distribution was used as it shown to successfully represent daily streamflow probability distribution (not shown). The DOY of peak daily annual streamflow and the DOY of streamflow's centre of mass decreased by -1.8 and 1.2 days decade$^{-1}$, respectively. This finding is consistent with the earlier snow depletion date shown in Figure **7**. The abnormally high value for the DOY peak daily annual streamflow and of streamflow's centre of mass (Figure 9c and d; DOY = 226, mid-August) for the water year 1968 is associated with a water year with abnormally high rainfall-runoff compared to snowmelt runoff. No trends were found in monthly streamflow volumes for each month between May and October (not shown), except for September, which decreased at about -47.1 m$^3$ decade$^{-1}$.

Figure 10 presents the mean daily streamflow discharge for observed streamflow (1995-2015) and the three modelling scenarios over the period 1960-2016. The ΔC and ΔCV scenarios show very similar mean hydrographs; with streamflow discharge starting in mid-April reaching the peak discharge at 0.7 m$^3$ s$^{-1}$ in June 8 and ending by mid-November. The ΔV scenario presents a much different mean discharge response, which is not surprising as meteorological drivers largely control the mean conditions and these were kept constant in this scenario. Under this scenario (ΔV), streamflow starts in mid-May reaching the peak discharge at 1.7 m$^3$ s$^{-1}$ in May 22, and it ends in mid-August, having a much shorter discharge season. The current mean hydrological regime, discussed in detail by Krogh et al. (2017), shows an earlier peak flow compared with the long-term ΔC or ΔCV scenarios, which is consistent with the reduction in the date of peakflow presented in Figure 9c. Also, larger late-fall streamflow discharge is present under current conditions.

### 5.5    Teleconnections

Table 5 lists Pearson correlations coefficients between annual basin scale mass fluxes and five climatic indices. Statistically significant correlation coefficients with p-values ≤ 0.05 are in bold. Significant correlations were found between some mass fluxes and North Pacific Index (NPI), Southern Oscillation Index (SOI) and Pacific Decadal Oscillation (PDO); however, even significant Pearson coefficients were relatively low (≤0.41), suggesting that large-scale climatic oscillations do not have an important effect on Havikpak Creek Basin hydrology. The same analysis on a seasonal scale provided similarly low correlation coefficients (not shown).

## 6    Discussion

### 6.1    Changing Meteorology

The increasing air temperature trends at Inuvik found in this study (Figure 5) qualitatively agree with those trends found by other studies using gridded data products (DeBeer et al., 2016; Vincent et al., 2015). Inuvik winters have warmed to the greatest degree; minimum and mean air temperature have increased by 8.0 and 5.2 °C, respectively, over 1960-2016. No temporal trend in precipitation was found at Inuvik (Figure 4), except for decrease in the spring (-2.7 mm decade$^{-1}$; Figure 4); however, the change point analysis showed an important decrease in year 1972 from 369 to 321 mm yr$^{-1}$ for the mean annual precipitation (Table 3). Vincent *et al.* (2015) investigated long-term trends in precipitation records over Canada for the period between 1948 and 2012 using the gridded and spatially interpolated CANGRD dataset (Rapaic *et al.*, 2015). For the region around Havikpak Creek Vincent *et al.* (2015) showed significant spatial variability with an small increase of less than 10% in annual precipitation. The CANGRD dataset is an spatially interpolated 50 km product that is based on the AHCCD dataset and has shown problems when compared against weather station data, particularly north of 60° N (Milewska and Hogg, 2001). Different trends found in this study and Vincent *et al.* (2015) can be explained by interpolation errors in the CANGRD dataset and the different period of analysis. This suggests that careful assessment of regional climate product needs to be performed when looking at individual sites, particularly in the Arctic where there are few stations.

As presented in section 3.1, the precipitation time series was produced using mostly the AHCCD dataset and corrected records from automated weather stations (AWS) for wind undercatch, producing a discontinuity in the time series in the mid-90s. Although uncertainty exists in the precipitation records, there is a relatively high confidence in the accuracy of precipitation, supported by the typically low wind speed limiting wind undercatch losses, the meticulous quality control and corrections used in the AHCCD dataset (Mekis and Vincent, 2011), the well-established wind-undercatch correction used for the AWS snow gauge, and the snow surveys from small clearing with minimal snow distribution and sublimation that allows a good estimation of seasonal snowfall. Comparing this precipitation dataset with another nearby station is challenging, as it there is no station with similar long-term records close to Inuvik. Nevertheless, the impacts of such uncertainty on the presented results are expected to be small and should not change the core discussions and conclusions of this study.

Mean annual short-wave irradiance from combined ERA-40 and ERA-I decreased by -1.4 W m$^{-2}$ decade$^{-1}$ (Table 2) or -7.4% over 1960-2016 with respect to 1960. Other studies have also found that measured solar irradiance in the Arctic has decreased. For example, Weston *et al.* (2007) found a decreasing trend in solar irradiance at two Canadian Arctic sites: Alert and Resolute Bay, Nunavut Territory, for the period 1964-2002 and 1957-2003, respectively. They argued that decreases in short-wave irradiance are driven by changes in atmospheric composition, such as aerosols and greenhouse gases, producing a decreasing in the calculated daily Clearness Index. However, ERA-I irradiance model calculation (Saunders et al., 1999) does not include the effect of aerosols scattering, but it does include the effect of greenhouse gasses, such as water vapour and carbon monoxide. Conversely, mean annual long-wave irradiance shows an increasing trend of 2.9 W m$^{-2}$ decade$^{-1}$ (Table 2) or 7.3% over 1960-2016 with

respect to 1960. This result agrees with global observations showing an increase in long-wave radiation (Ohmura, 2009), particularly over the Canadian Arctic, for which observed net long-wave is also increasing (Weston et al., 2007), and is consistent with an increase in cloud cover and/or water vapour in the atmosphere with resulting increasing atmospheric emissivity and/or increasing air temperatures. The annual modelled all-wave irradiance is increasing by 2.6%, but with seasonal variations. Winter all-wave irradiance has been increasing by 10% providing more energy to snowmelt and sublimation, whereas summer all-wave irradiance has been decreasing by 3%, which decreases the energy for ET and ground thaw.

## 6.2    Changes to the Hydrological Cycle

Precipitation phase shifted from snowfall to rainfall in the scenarios including climate change ($\Delta C$ and $\Delta CV$; Figure 8) by 22.7% from 1960 to 2016, this was driven by the increase in mean annual air temperature of 3.7 °C (Table 1). Snowcover duration decreased for $\Delta C$ (some HRUs; Figure 7), whereas for $\Delta V$ both decreased and increased over the HRUs; however, the $\Delta CV$ resulted in a shortened snow season (most HRUs). This shortening was mostly driven by changing climate with reduced snowfall and snow redistribution to drifts by wind. Similarly to the snow season duration, the snowcover depletion date for $\Delta CV$ decreased between 8 and 17 days over 1960-2016 (Figure 7), with the greatest decrease in the Upper Gully/Drift HRU, due to decreasing blowing snow redistribution and hence peak SWE. As peak streamflow in HPC is dominated by snowmelt events, these changes are consistent with the 10 day advance in peak streamflow date. Peak SWE decreased between 12 and 33% in the $\Delta C$, whereas for $\Delta V$ it increased in the Sparse Shrub HRUs by 3 to 30% and decreased in the Gully/Drift HRUs by 22 to 40%, respectively. The $\Delta CV$ scenario resulted in diminishing peak SWE by 12 to 50%, due to the combination of decreasing snowfall and blowing snow redistribution from Sparse Shrubs to Gully/Drift HRUs. Snow ablation rates for $\Delta C$ decreased by 0.3 to 1.1% over 1960-2016, whereas for $\Delta V$ decreased in the Sparse Shrub HRUs by 0.1 to 0.4% and increases by 0.31 to 0.4% in the Gully/Drift HRUs. Changes in snow ablation rates due to a warmer climate have been investigated in other cold regions. Rasouli *et al.* (2014) and Pomeroy et al. (2015) modelled snow hydrology in mountain basins in Yukon and Alberta, Canada, respectively, and attributed the lower snow ablation rates under climate change to an earlier snowmelt season, occurring when lower solar radiation inputs are available. Using snow accumulation records in western U.S.A., Musselman *et al.* (2017) reached a similar conclusion. López-Moreno et al. (2012) also found a reduction in ablation rates in the Spanish Pyrenees under a scenario of warmer temperatures. However, here some snow ablation rates increased for $\Delta V$, suggesting climatic factor are not the only control in ablation rates, but that vegetation dynamics can compensate or even reverse trends in ablation rates due to changing climate.

Sublimation decreased in $\Delta C$ by 23%, due to a decrease in sublimation of intercepted snow by 19%. Factors decreasing sublimation of intercepted snow are warmer temperatures, causing accelerated snow unloading from the canopy, and decreasing snowfall. Sublimation for $\Delta V$ decreased by 10%, due to blowing snow sublimation dropping by 44%, mostly in the upper basin. Decreasing blowing snow in this scenario is driven by shrub densification, increasing the aerodynamic roughness height and wind speeds required to initiate blowing snow transport. Shrub densification intensified the effect of changing climate on sublimation, decreasing sublimation by 29% over the study period. ET was found not to change in the changing climate-only scenario; however, the ET of intercepted

rainfall and soil moisture-restricted ET from the P-M or P-T equations decreased by 51 and 6%, respectively. This is explained by the different inter-annual variability of the two ET terms hampering the individual trends; nevertheless, the change point analysis of mean annual ET shows a decrease of 16 mm yr$^{-1}$ in 1977 (9.5% with respect to 1960 for ΔC and ΔCV), consistently with the simulated trends of each ET component. ET for ΔV increased by a marginal 0.2%, due to the 1.5% increase in soil moisture content and 0.8% increase in ET of intercepted rainfall due to shrub expansion. The combined effect of changing climate and vegetation decreased ET by 8.5%, driven largely by changing climate. Decreasing summer all-wave irradiance (3%) and soil moisture content (19%) were driving decreasing ET in the scenario with combined climate and vegetation changes.

Active layer thickness (ALT) for ΔC increased by 11 to 28 cm over 1960-2016 for most HRUs, caused by the earlier snow depletion date (8 to 11 days) and ground thaw initiation (6 to 11 days), and warmer ground-surface temperatures due to warmer air temperatures. ALT increased up to 6 cm for ΔV in some HRUs, driven by the earlier snowcover depletion date (3 to 8) and ground thaw initiation (2 to 6 days). The effect of changing vegetation dampened the deepening in ALT found in ΔC scenario for most HRUs; nevertheless, in the ΔCV scenario, ALT increased by 11 to 22 cm.

Annual Streamflow volume from HPC has dropped by 38 mm (21%) whilst annual precipitation has dropped by 48 (13%) mm since 1960. We argue that the 10 mm (21%) difference between the decrease in precipitation and streamflow discharge from HPC suggest a small degree of hydrological resiliency, here defined as the capacity of a basin to actively counteract the impact of changes in weather on streamflow discharge, which is explained by the declining ET and sublimation. This result emphasizes the need for a full physically based representation of the hydrological cycle in models so that the processes driving this resiliency can be used to diagnose its function.

### 6.3    Havikpak Creek Basin changes versus other Arctic studies

The ΔCV scenario best represents historical change in Havikpak Creek Basin; therefore, it is used to compare with other Arctic studies. Snowcover depletion dates in HPC accelerated between 1.5 and 3.2 days decade$^{-1}$ (Figure 7), which are higher than the average trend presented by Liston and Hiemstra (2011) for the entire Arctic (-1.28 days decade$^{-1}$), but smaller than their largest trend in the Arctic (-9.89 days decade$^{-1}$). The maximum ALT depth increased by 1.8 to 4.2 cm decade$^{-1}$ (Figure 7), which is smaller than the average trend of 4.7 cm decade$^{-1}$ modelled by Oelke *et al.* (2004) over the Mackenzie River Basin. Differences in ALT change simulations can be due to: (1) differences in the model's spatial representation, Oelke et al. used grids of 25 km, with which small-scale features are not well represented; (2) differences in the ground freeze/thaw method algorithm, Oelke et al. use a one-dimensional heat conduction (i.e. lateral flow is neglected); and (3) the driving meteorology used by Oelke et al. was the NCEP/NCAR reanalysis, which has shown some problems in representing Arctic climate (Serreze et al., 1998; Serreze and Hurst, 2000). However the average permafrost conditions of the Mackenzie River Basin (MRB) are thinner and warmer compared with those in HPC, and so average changes in ALT are expected to be larger for the MRB than for HPC.

Annual streamflow volume at HPC has dropped (Table 4); unfortunately there are no long-term studies of small streams that originate in the Arctic to compare this result with. There are studies showing increasing large river basin streamflow into the Arctic (McClelland et al., 2006; Overeem and Syvitski, 2010; Peterson et al., 2002; Rood et al., 2017; Yang et al., 2002). However, a significant portion of the runoff in these basins originates south of the Arctic Circle (e.g. the Mackenzie and the Lena River basins in Canada and Russia, respectively), and therefore these trends are not representative of changes in Arctic hydrology. Previous studies have argued that the increase in the streamflow of large rivers flowing into the Arctic is driven by increasing baseflow due to permafrost thaw and increasing precipitation. However, HPC annual precipitation and streamflow have both dropped and the earlier shifts in the hydrograph are inconsistent with such mechanism. Instead, baseflow during the end of the summer is minimal, streamflow has been decreasing during September, and no winter flow has been observed. Only a few similarities can be found between results of studies of large river basins flowing to the Arctic and HPC, such increasing ALT and accelerating snow-free date; however, most processes, such as evapotranspiration and streamflow depend on the local scale interaction between several physical processes, which are undergoing distinct changes that are not evident in rivers flowing into the Arctic. Therefore, the results of studies of these large river basins should not be confused with the results for an Arctic hydrology study.

This study considered changing climate and transient vegetation change separately to identify their individual effects; nevertheless, they are strongly coupled in the historical record. Warming temperatures are well correlated with shrub growth (Myers-Smith et al., 2011), which has a positive feedback to atmospheric heating by decreasing surface albedo, generating greater sensible heat flux to the atmosphere (Pomeroy et al., 2006) and a negative feedback by consuming more atmospheric $CO_2$ (Myers-Smith et al., 2011). The modelling scenario experiments here revealed that most simulated trends in the water balance are attributable to changes in climate; however, the effect of transient vegetation as expressed in shrub expansion and densification, was shown to further reduce blowing snow redistribution and sublimation, which intensified climate change-driven trends produced by the reduced snow accumulation. This emphasizes the need to included transient vegetation changes in hydrological simulations, which is typically neglected in hydrological models. Reliable rates of change in vegetation species, height and density need to be available for this purpose; therefore, comprehensive studies investigating these changes in other transitioning environments are needed.

## 7    Conclusion

This study diagnosed changes in the hydrology of a small Arctic basin in the tundra-taiga transition using a spatially distributed and physically based hydrological model. It considered both transient climate and vegetation changes for the first time. There was no evidence for intensification of the hydrological cycle as instead, most processes slowed. In the changing climate-only scenario, statistically significant changes were found for diminishing snow accumulation, sublimation, blowing snow redistribution, snowcover duration, snow ablation rate, and evapotranspiration, deepening active layer thickness, and earlier snowcover depletion and ground thaw initiation. These, along with warming temperatures, declining summer net radiation and declining precipitation, resulted in

diminished annual streamflow volume of 40 mm over the 56 years. However the decline in streamflow did not match the larger decline in precipitation (48 mm), providing some evidence of resilience to climate change, as despite rising temperatures, both evapotranspiration and sublimation dropped with declining precipitation and this attenuated the streamflow volume decline. Transient vegetation change further decreased blowing snow sublimation by reducing blowing snow transport. The combination of changing climate and transient vegetation change resulted in annual streamflow volume dropping by 38 mm over 56 years – a change that is not substantially different from that due to climate change alone. These results suggest that historical changes in vegetation and a degree of hydrological resiliency have not compensated for the effects of climate change on the hydrological regime of Havikpak Creek. They provide the first estimates of long-term change for a drainage basin located completely within the Arctic Circle, and demonstrate the large, complex and recent hydrological changes that have occurred, which can be used as a reference to inform other studies of Arctic climate change impacts.

*Author contribution.* SK and JP designed the study. JP developed the CRHM modelling platform. SK performed the simulations, the statistical analyses, and prepared the manuscript with contributions from JP in the manuscript structure, readability and analysis and discussion of the results.

*Competing interests.* The authors declare that they have no conflict of interest.

*Acknowledgments.* Financial support for this study was provided by CONYCIT under the BECAS CHILE scholarship program, Global Water Futures, Changing Cold Regions Network, NSERC Discovery Grants, Canada Research Chairs and Yukon Environment. The authors thank Tom Brown of the Centre for Hydrology for providing technical support with the CRHM platform. Comments from Joseph Shea, Paul Whitfield, Lucia Scaff, and two anonymous reviewers are greatly appreciated.

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

**Tables**

Table 1: Changes in precipitation and temperature for the period 1960-2016 and statistically significant trends at the p≤0.05 significance threshold using Mann-Kendall test. Changes in precipitation as percentage with respect to 1960 are also presented.

| Period | Precipitation (mm) | Minimum Air Temperature (°C) | Mean Air Temperature (°C) | Maximum Air Temperature (°C) |
|--------|--------------------|-----------------------------|---------------------------|------------------------------|
| Winter | - | 8.0 | 5.2 | - |
| Spring | -15.1 (27%) | - | 2.7 | - |
| Summer | - | - | 0.8 | 1.8 |
| Fall | - | - | 1.6 | - |
| Annual | - | 8.0 | 3.7 | 1.8 |

Table 2: Slope for statistically significant weather trends at the p≤0.05 significance threshold using Mann-Kendall test.

| Period | Short-wave Irradiance (W m$^{-2}$ decade$^{-1}$) | Long-wave Irradiance (W m$^{-2}$ decade$^{-1}$) | All-wave Irradiance (W m$^{-2}$ decade$^{-1}$) | Wind speed (m s$^{-1}$ decade$^{-1}$) | Relative Humidity (% decade$^{-1}$) |
|--------|------------------|-----------------|----------------|------------|-------------------|
| Winter | 0.8 | 2.8 | 3.9 | - | 0.6 |
| Spring | -4.3 | 2.4 | - | - | - |
| Summer | -6 | 3.3 | -2.9 | -0.1 | 1.4 |
| Fall | - | 3.7 | 2.4 | - | 1.2 |
| Annual | -1.4 | 2.9 | 1.5 | - | 0.8 |

Table 3: Mean change point analysis of the atmospheric forcing variables.

| Atmospheric Variable | Mean Annual Change | Year |
|----------------------|--------------------|------|
| Precipitation (mm) | 369 to 321 | 1972 |
| Air Temperature (°C) | -9.1 to -7.1 | 1991 |
| Short-wave Irradiance (W m$^{-2}$) | 112 to 104 | 1969 |
| Long-wave Irradiance (W m$^{-2}$) | 230 to 242 | 1969 |
| All-wave Irradiance (W m$^{-2}$) | 344 to 348 | 1997 |
| Wind Speed (m s$^{-1}$) | N/A | N/A |
| Relative Humidity (%) | 69 to 75 | 2013 |

**Table 4: Change point analysis for selected annual basin-scale mass fluxes for the three modelling scenario**

| Mass Fluxes | ΔC: Δ Climate-only | | ΔV: Δ Vegetation-only | | ΔCV: Δ Climate and Vegetation | |
|---|---|---|---|---|---|---|
| | Mean Change (mm) | Year | Mean change (mm) | Year | Mean Change (mm) | Year |
| Rainfall | 131 to 196 | 2013 | N/A | N/A | 131 to 196 | 2013 |
| Snowfall | 211 to 169 | 1997 | N/A | N/A | 211 to 169 | 1997 |
| Sublimation | 39 to 28 | 2013 | 37 to 35 | 1988 | 42 to 36 | 1980 |
| ET | 160 to 144 | 1977 | N/A | N/A | 160 to 144 | 1977 |
| Soil Moisture | 80 to 48 | 1968 | N/A | N/A | 82 to 49 | 1968 |
| Streamflow | 180 to 140 | 1973 | 133 to 135 | 1992 | 178 to 140 | 1973 |

**Table 5: Pearson correlation coefficient between basin-scale mass fluxes and climatic indexes, using water year values (October-September). Correlation coefficients with p-value ≤0.05 are in bold.**

| Climatic Index | AO | NAO | NPI | SOI | PDO |
|---|---|---|---|---|---|
| **Rainfall** | 0.134 | 0.002 | 0.207 | -0.007 | -0.110 |
| **Snowfall** | -0.013 | 0.151 | 0.116 | -0.168 | 0.022 |
| **Precipitation** | 0.075 | 0.114 | 0.219 | -0.130 | -0.054 |
| **Sublimation** | -0.021 | 0.125 | 0.256 | 0.185 | **-0.340** |
| **Blowing Snow Sublimation** | 0.044 | 0.020 | **0.336** | 0.200 | **-0.397** |
| **Snowpack Sublimation** | -0.049 | 0.077 | 0.108 | 0.203 | -0.249 |
| **Sublimation of Intercepted Snowfall** | -0.014 | 0.172 | 0.143 | -0.071 | -0.058 |
| **Restricted ET from P-M or P-T equations** | 0.161 | -0.200 | 0.253 | **0.268** | **-0.332** |
| **Evaporation from Canopy Interception** | -0.034 | 0.125 | -0.017 | -0.078 | 0.061 |
| **ET** | 0.156 | -0.183 | 0.250 | 0.256 | **-0.323** |
| **Soil Moisture** | -0.010 | -0.158 | 0.191 | -0.065 | -0.073 |
| **Streamflow** | -0.002 | 0.062 | 0.083 | -0.262 | 0.141 |

**Figures**

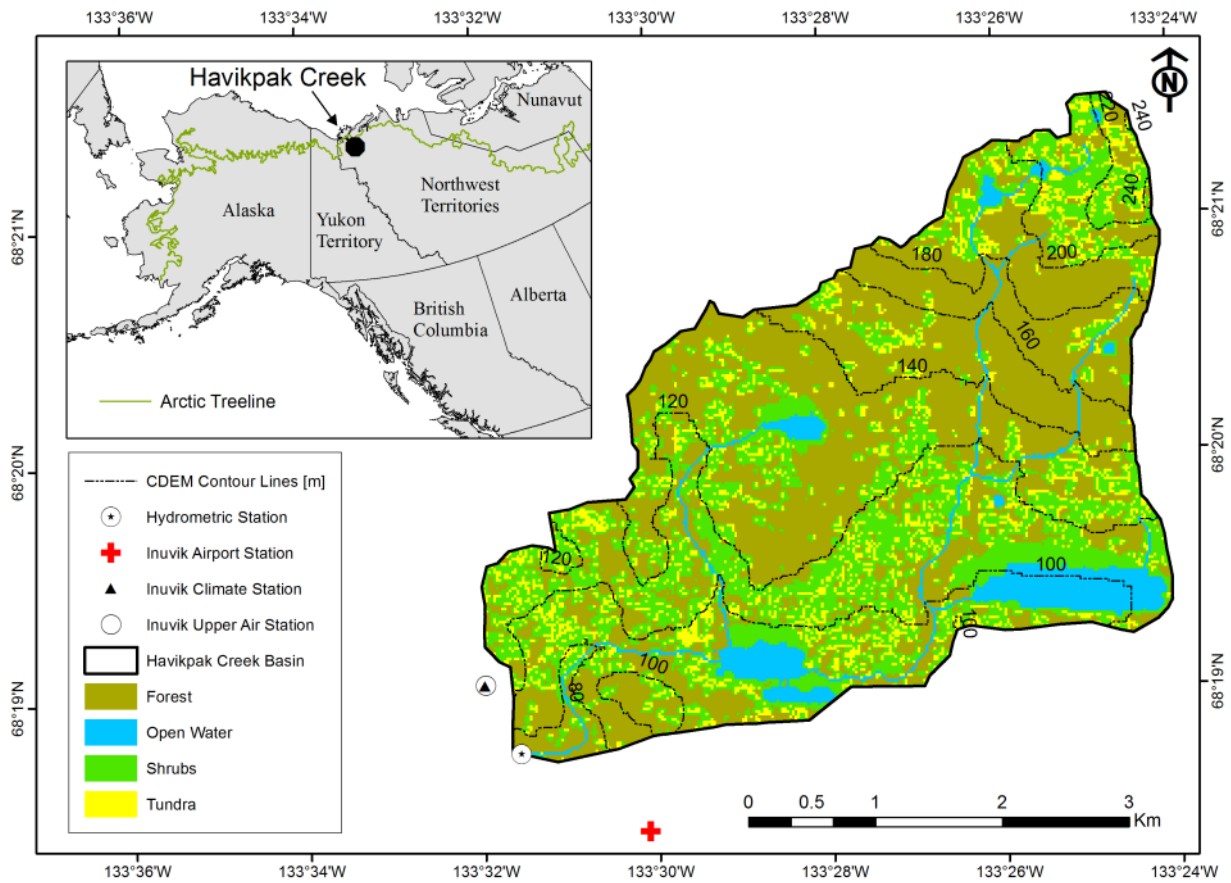

**Figure 1: Havikpak Creek Basin including elevation contour lines (based on the Canadian DEM – 20m), the location of weather and hydrometric stations, and the 1992 landcover map based on Krogh *et al.* (2017). Inset plot shows the location of the study site within North America and the approximate location of the Arctic treeline.**

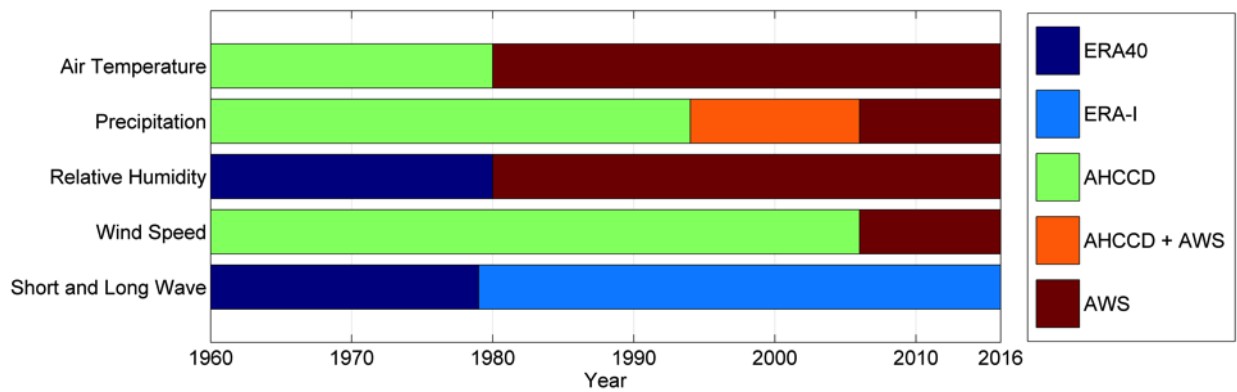

**Figure 2: Data source for each of the weather variables during the period 1960-2016. AWS: Automatic Weather Stations. AHCCD: Adjusted and Homogenized Canadian Climate Data. ERA-I: ERA-Interim.**

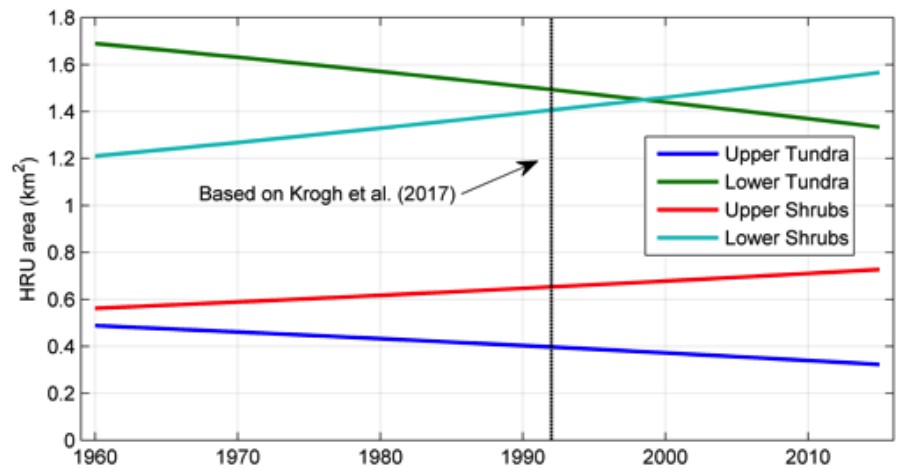

**Figure 3: Annual changes in the Tundra and Shrubs HRUs area used in the CRHM-AHM model.**

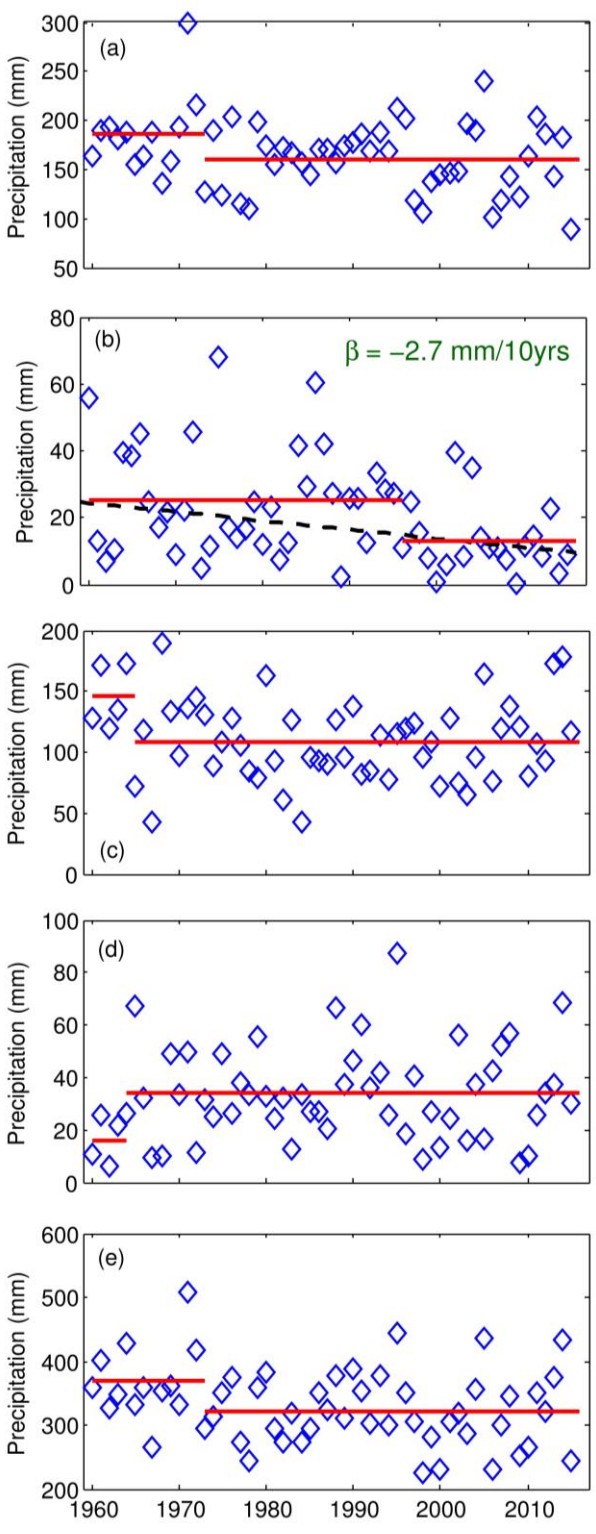

**Figure 4: Observed seasonal and annual precipitation for each water year (October-September) from 1960 to 2015. (a) Winter (Oct-Apr), (b) spring (May), (c) summer (Jun-Aug), (d) fall (Sep) and (e) annual. Slope (β) is shown in mm decade[-1] for statistically significant trends at the p≤0.05 significance threshold. Solid red line shows the annual change point.**

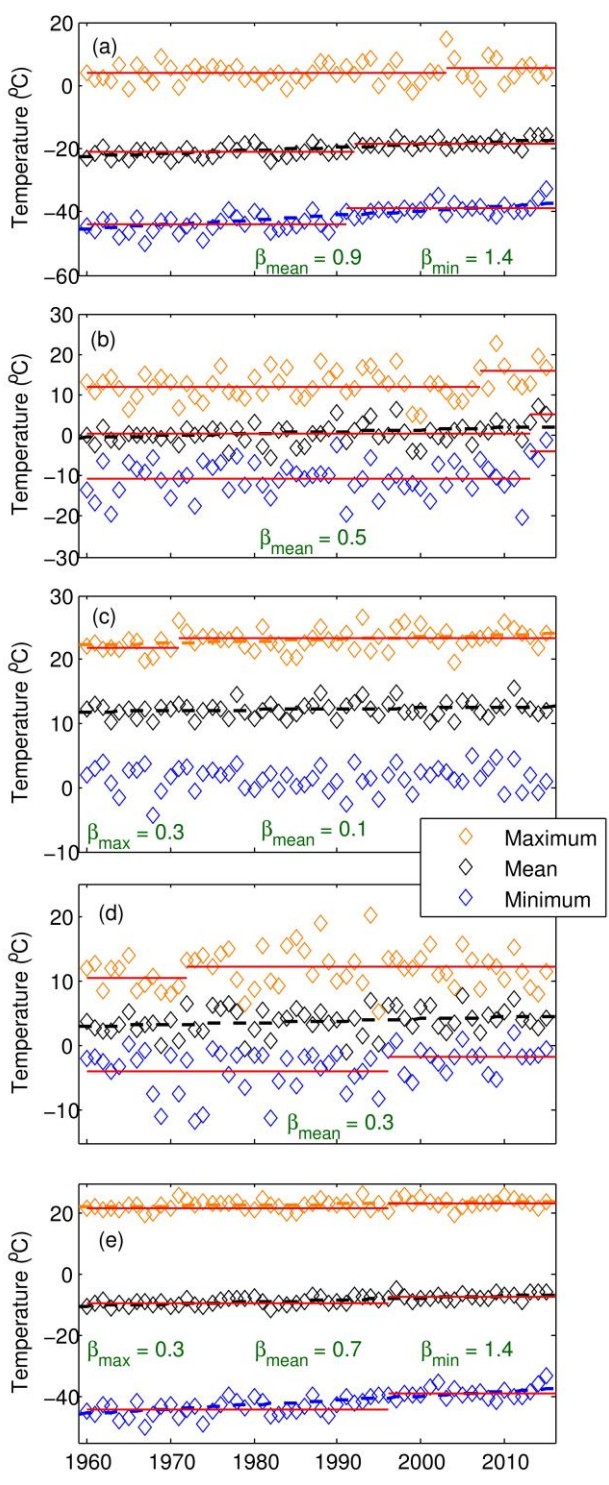

**Figure 5: Observed seasonal and annual maximum, mean and minimum temperature for each water year (October-September) calculated from mean daily temperature, between 1960 and 2015. (a) Winter (Oct-Apr), (b) spring (May), (c) summer (Jun-Aug), (d) fall (Sep) and (e) annual. The dashed line is the linear regression using Sen (1968). Slope (β) in °C decade$^{-1}$ for statistically significant trends at the p≤0.05 significance threshold is shown. Solid red line shows the annual change point.**

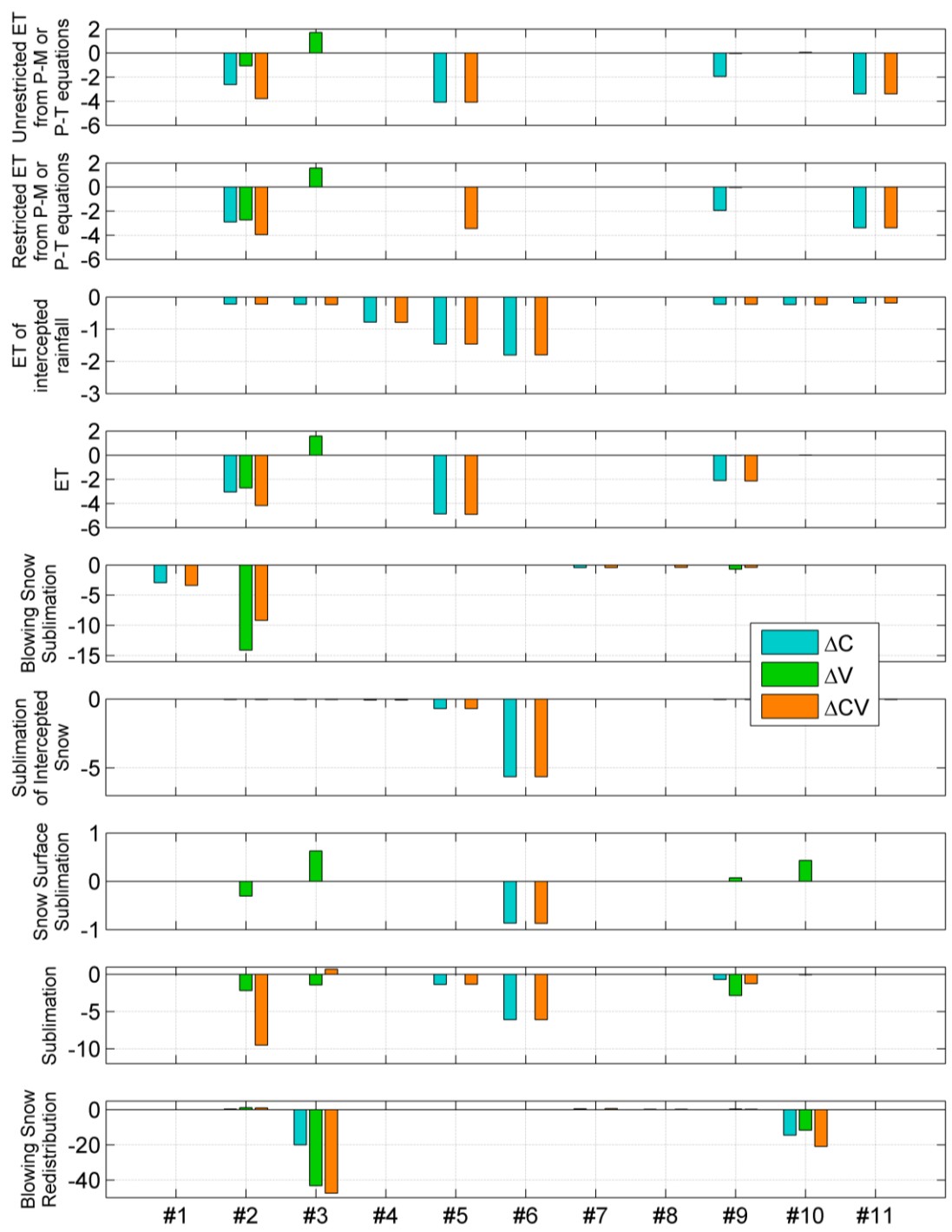

**Figure 6: Units in mm decade$^{-1}$.Scenario comparison of significant trends (p≤0.05) for selected mass fluxes at an HRU-scale. X-axis as follows: Upper Tundra (#1), Upper Sparse Shrubs (#2), Upper Gully-Drift (#3), Close Shrubs (#4), Taiga Forest (#5), Forest (#6), Lower Tundra (#7), Open Water (#8), Lower Sparse Shrubs (#9), Lower Gully-Drift (#10) and Wetland (#11).**

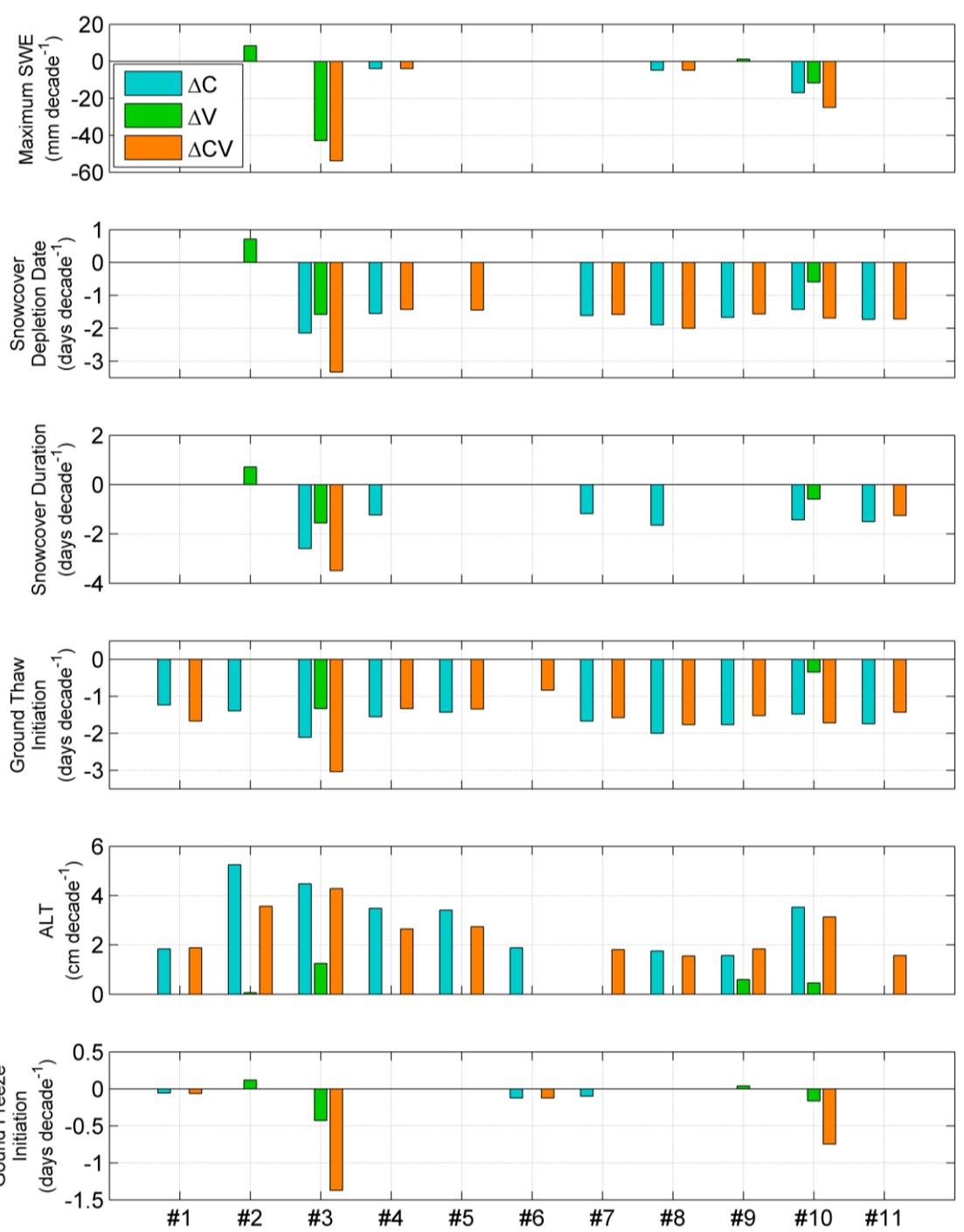

**Figure 7: Comparison of significant trends (p≤0.05) for snow and ground freeze/thaw related variables at an HRU-scale for the three scenarios. Note that trends for snowcover depletion date, snowcover duration and ground thaw initiation are in dates, and for maximum SWE, ALT and snow ablation are in rates. X-axis as follows: Upper Tundra (#1), Upper Sparse Shrubs (#2), Upper Gully-Drift (#3), Close Shrubs (#4), Taiga Forest (#5), Forest (#6), Lower Tundra (#7), Open Water (#8), Lower Sparse Shrubs (#9), Lower Gully-Drift (#10) and Wetland (#11). ALT: Active Layer Thickness. SWE: Snow Water Equivalent.**

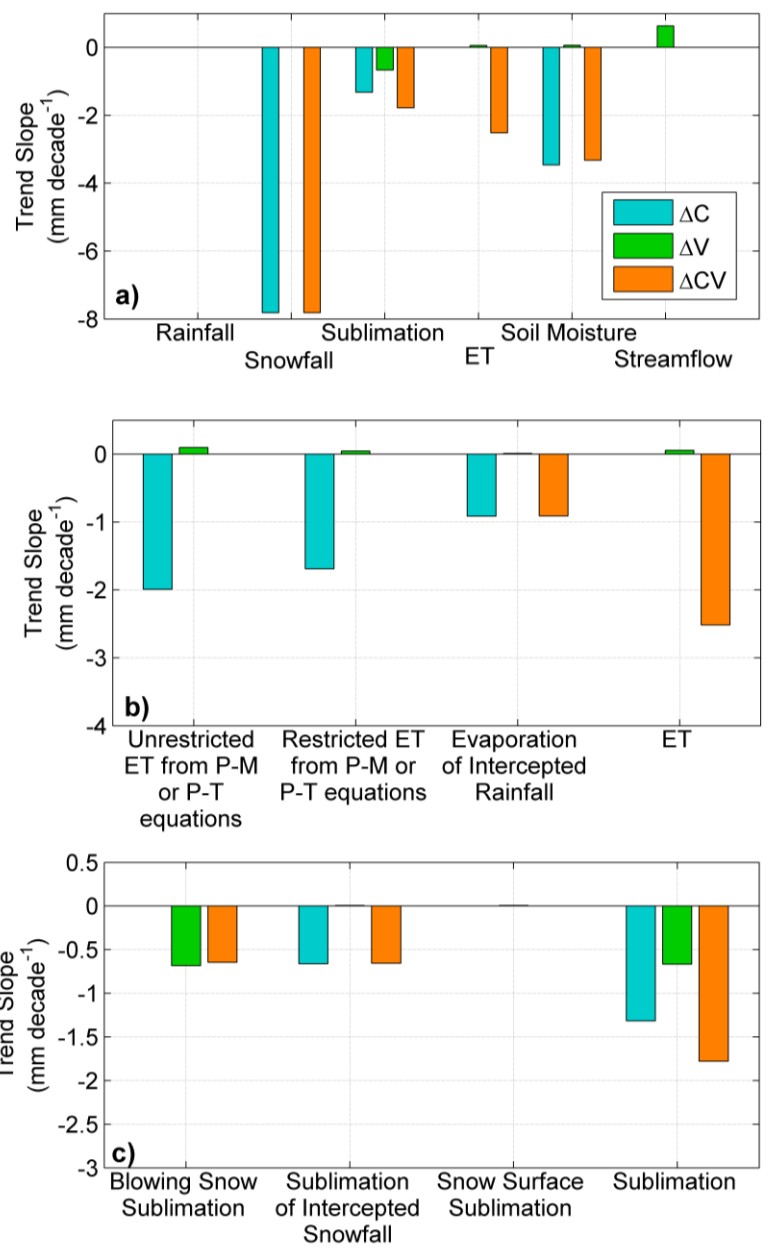

**Figure 8: Comparison of basin-scale annual mass fluxes trends (p≤0.05) over the water years from 1960 to 2015, for the three scenarios. a) Main mass fluxes. b) Evapotranspiration components. c) Sublimation components.**

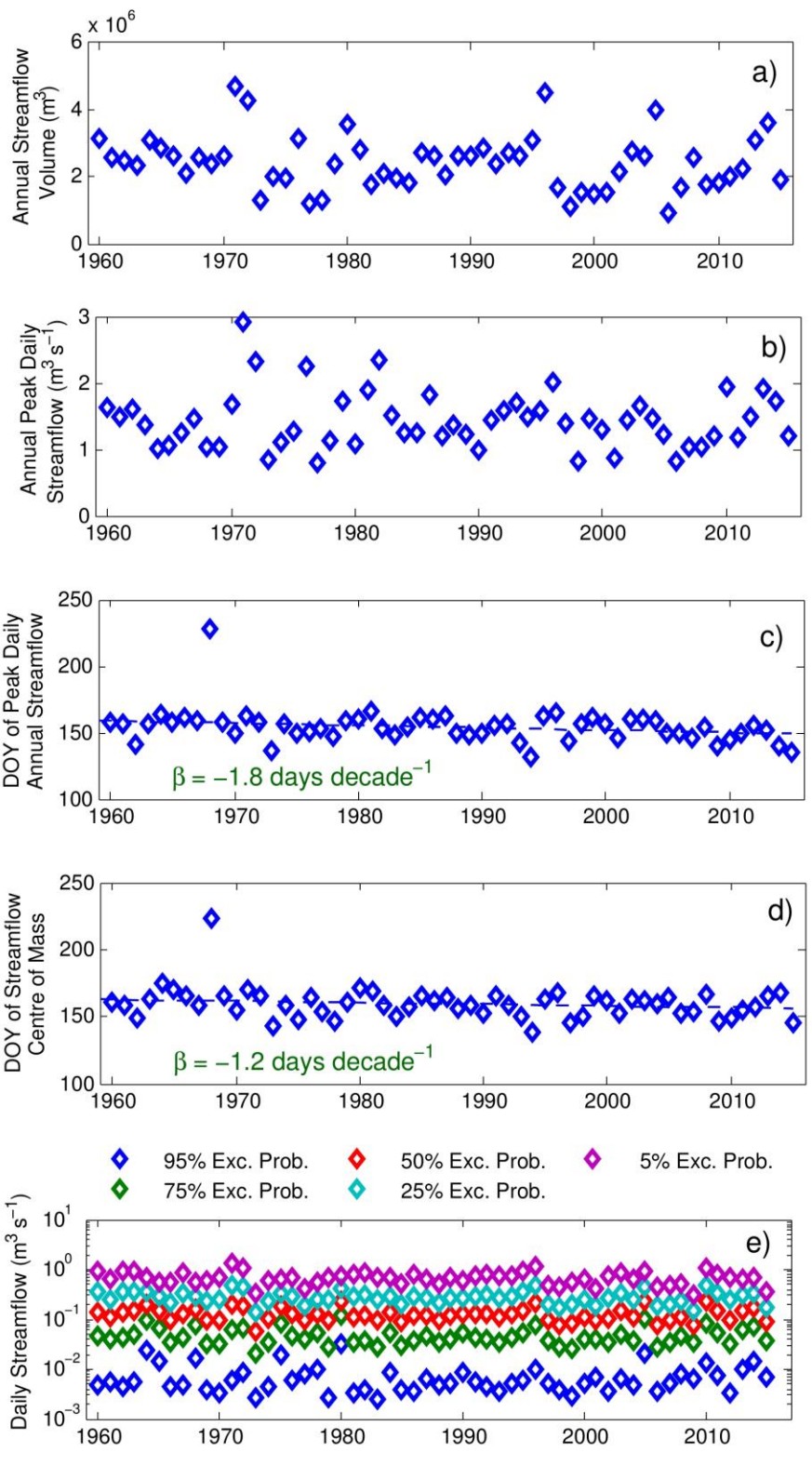

**Figure 9: a) Annual streamflow volume. b) Annual peak daily streamflow. c) Day of the Year (DOY) of peak daily streamflow. d) Day of the Year (DOY) of streamflow volume discharge centre of mass. e) Streamflow discharge associated for various exceedance probabilities. X-axis of all subplots is the water year starting in October.**

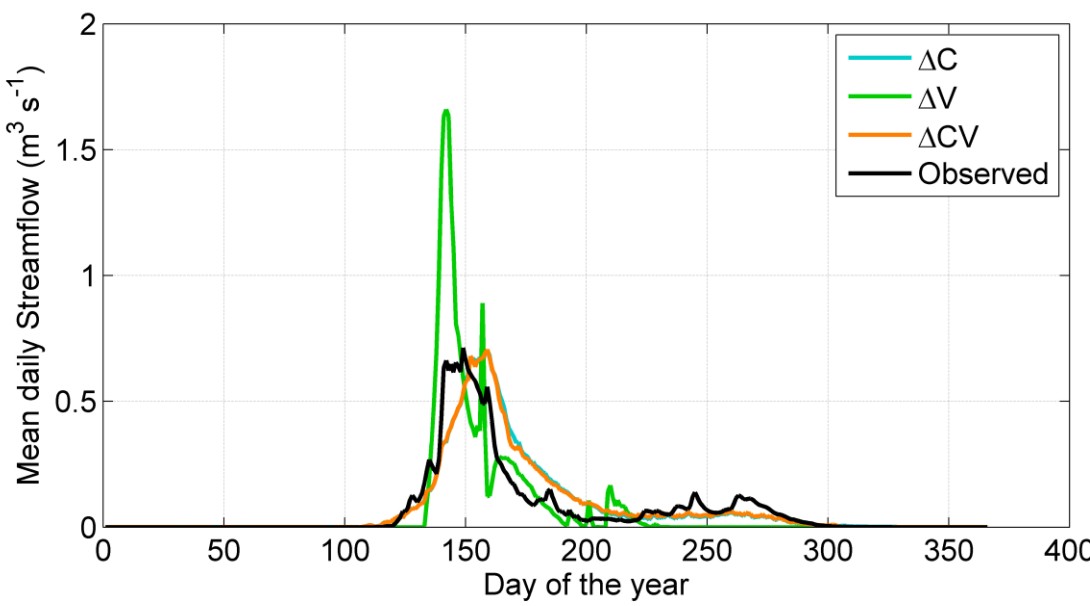

**Figure 10: Mean annual hydrograph for the observed streamflow (1995-2015), and the three modelling scenarios: changing climate-only (ΔC), changing vegetation-only (ΔV) and changing climate and vegetation (ΔCV). Note the overlapping between the ΔC and ΔCV scenarios.**