# Peer review of "Recent Changes to the Hydrological Cycle of an Arctic basin at the Tundra-Taiga Transition"

_Hydrology and Earth System Sciences, 2018_

## Referee Comment (RC1) · Anonymous Referee #1 · 21 Mar 2018

The paper shows the changes occurred in climate, vegetation and hydrology of a small basin located in the Canadian Arctic. This is one of first studies that make a very complex analysis and modeling of the global change impacts on the hydrology of an arctic catchments, and it is a nice contribution for a journal as HESS. The paper is well written and, in general, the methodology is clearly explained. The authors have need to combine different data sources in order to be able to analyze the changes in the long-term (going back to the 60′s). This can be the source of some uncertainty, but I also think that this is the only way possible to study global change processes in remote areas, and the main results should not be affected by the used data. Following, I enumerate a number of questions (mostly minor issues) that authors can consider for preparing a revised version of the paper. 1- In the abstract change is shown in absolute units, I think they can also be presented as a % to get a faster idea of the magnitude of the change. 2- When shrub expansion in the arctic is mentioned, I miss the reference to Sturm et al 2011. doi:10.1038/35079180. 3- Were vegetation changes determined by aerial photographies? I have not found this information. 4- I would like to see a very brief description of the method to disaggregate daily into hourly precipitation (even when references are provided). 5- What is the advantage of using a "normal year" compared to use detrended series of historical climate?. 6- It results a bit confusing the statement that there are not significant trends in seasonal precipitation, but all of them show break points. I think this is because the breakpoints in most of the cases occur at the beginning of the study period and it prevents affecting the trend analysis. Is this the reason? It could be mentioned in the discussion or when presenting the results. 7- Are the changes in DOY of peak discharge statistically significant? 1.5 days per decade do not seem a very big change. 8- A reduction in melt rates due to warmer temperatures was also presented by López-Moreno et al., 2012. DOI: 10.1002/hyp.9408 9- A figure showing the mean annual hydrograph for control, changing climate, changing vegetation and changing climate and vegetation might result illustrative. 10- When hydrological resilience is discussed in pag. 16, the use of change in P and Q in % (in addition to mm) would help. Probably the use of the term "mild resilience" is a bit unclear. 11- I would include the changes in P as % in Tables 1 and 2. 12- I would show al the trend values but just highlighting the statistically significant ones (as in Table 5) 12-Title of Table 5, change bolt by bold. 12- The Labels of the X-axis in Figure 8 overlap.

I hope my comments will result useful.

---

## Referee Comment (RC2) · Anonymous Referee #2 · 31 May 2018

I think this is a generally well-written and important paper that attempts to separate the drivers of climate change and vegetation change on hydrology over a historic period. My biggest concern is the observational data used to drive the model, particularly precipitation and snow. I understand that the experiment compares three scenarios (changing climate, changing vegetation, and both) so I would like to see how each of these is impacted by uncertainty in the quality of the precipitation estimates. If the cold season precipitation is most biased, it could be that changes in the length of the cold season cause changes in this bias with time. I think the springtime precipitation trend is suspicious and I'd like to know how uncertainty in those data impacts the robustness of the results. The description of the precip data is useful, but I'd have

more confidence if a more thorough comparison of different precip data sources (and nearest other stations) were performed. There are a lot of discontinuities in these datasets, as described. Have the authors considered doing a scenario that separates temperature and precipitation change? In the discussion, I'd like to hear more about impacts of the uncertainty in the precipitation data on the larger results of the study. Same for snow measurements and to a lesser degree, streamflow.

Minor comments: like the other reviewer, I don't care for the use of the term 'hydrologically resilient' without a technical definition provided. This is too vague. I would also like a little more information on this basin. Is this a well-instrumented research basin? It doesn't really seem like it, based on the description of the single station observations. Why was it chosen? Are there no research basins that fit the description (tundra-taiga boundary with permafrost)? In your introduction, it might be worth mentioning the NASA ABoVE (Arctic-Boreal) campaign, focused on exactly these eco-zones because it has a hydrology component. Finally, while most of the paper is readable, the abstract could use some work. Go for shorter, simpler sentences that really convey what is interesting and exciting about this paper.

---

## Editor Comment (EC1) · B. Schaefli (Editor) · 12 Jun 2018

Both reviewers agree that this is a good to excellent manuscript and that it requires only minor revisions before it can be published. I invite the authors to submit a revised version that includes all the modifications discussed in the online discussion.

---

## Author Response (AR1)

Dear Editor,

Please find attached the reviewed version of the manuscript with and without track-changes, and the point-by-point response to the reviewer's comments. Main changes to the manuscript include:

- Abstract (Page 1)
- Reasons why Havikpak Creek was selected as the study basin (Page 4)
- Discussion of Figure 10 (Page 14)
- Discussion about precipitation uncertainty (Page 15)
- Definition of "Resiliency" (Page 17)
10
- The inclusion of Figure 10 (Page 40)

I hope you consider that we properly addressed all the reviewer's comments and that our manuscript fulfills HESS publishing standards.

Best Regards,

Sebastian Krogh

Response to RC1 from Krogh and Pomeroy

We appreciate the thoughtful comments and insights provided by Reviewer#1, and below detail a response to each comment.  **Responses are in bold**.

1- In the abstract change is shown in absolute units, I think they can also be presented as a % to get a faster idea of the magnitude of the change.

**Response: We changed the abstract as per reviewer#2 suggestions (shorter and simpler sentences) to make it more readable and opted to remove the details with the trend numbers.**

2- When shrub expansion in the arctic is mentioned, I miss the reference to Sturm et al 2011. doi:10.1038/35079180.

**Response: Yes, this is an important paper and it has been added to the revised version of the manuscript.**

3- Were vegetation changes determined by aerial photographies? I have not found this information.

**Response: Yes, vegetation changes were determined by Lantz et al (2013) using air photos.**

 4- I would like to see a very brief description of the method to disaggregate daily into hourly precipitation (even when references are provided).

**Response: A brief description of the microcanonical cascade model used to disaggregate precipitation has been included in the revised version of the manuscript as per the reviewer's suggestion.**

 5- What is the advantage of using a "normal year" compared to use detrended series of historical climate?.

**Response: The main advantage of using a "normal year" is that any trend or change point will be strictly associated with changes in vegetation, whereas in using a detrended weather series, changes in interannual variability may result in trends or change points not strictly associated with vegetation change.**

6- It results a bit confusing the statement that there are not significant trends in seasonal precipitation, but all of them show break points. I think this is because the breakpoints in most of the cases occur at the beginning of the study period and it prevents affecting the trend analysis. Is this the reason? It could be mentioned in the discussion or when presenting the results.

5 **Response: It is true that only spring precipitation shows a statistically significant and decreasing trend, whereas all the seasons show a decreasing change point. These two statistical techniques were used as they complement each other; however, a change in the mean does not necessarily produce a significant trend and vice versa. The reviewer's suggestion is reasonable as change points near the beginning or end of the time series will**
10 **more likely result in no-trend, although this also depends on the selected significance threshold. This discussion has been added to the revised version of the manuscript.**

7- Are the changes in DOY of peak discharge statistically significant? 1.5 days per decade do not seem a very big change.

**Response: Yes, the trend is statistically significant. Throughout the manuscript only trends**
15 **that are statistically significant at p-values <= 0.05 are presented. Although this trend might seem small (in fact it is -1.8 days/decade, figure 9c), over 60 years it results in peak flows occurring 10.8 days earlier.**

8- A reduction in melt rates due to warmer temperatures was also presented by López-Moreno et al., 2012. DOI: 10.1002/hyp.9408

20 **Response: Yes, an excellent paper. We have included this reference as per reviewer's suggestion.**

9- A figure showing the mean annual hydrograph for control, changing climate, changing vegetation and changing climate and vegetation might result illustrative.

**Response: We agree that such figure can be illustrative when looking at the mean**
25 **hydrological conditions at Havikpak Creek and how they will change over time and so have included a figure showing the mean hydrograph under control, changing vegetation alone, changing climate alone, and changing vegetation and climate. This is referenced to Krogh et al's (2017) detailed analysis of the mean hydrological regime of Havikpak Creek.**

10- When hydrological resilience is discussed in pag. 16, the use of change in P and Q in % (in addition to mm) would help. Probably the use of the term "mild resilience" is a bit unclear.

**Response: Yes, change in % has been added to the revised version of the manuscript. We have modified and clarified the use of the term hydrological resilience in the paper.**

11- I would include the changes in P as % in Tables 1 and 2.

**Response: Change in % has been included for precipitation in Table 1 in the revised version of the manuscript. Table 2 does not include precipitation.**

12- I would show al the trend values but just highlighting the statistically significant ones (as in Table 5)

**Response: We do not want to suggest trends where they are not statistically significant. Our argument here is that a non-significant trend is not a trend. As such, we would like to keep it as it is, to avoid confusion between slopes with non-significant and significant trends during the analysis and discussion and have not changed this.**

13-Title of Table 5, change bolt by bold.

**Response: This has been changed as per reviewer's suggestion.**

14- The Labels of the X-axis in Figure 8 overlap.

**Response: This has been fixed as per reviewer's suggestion.**

**Krogh and Pomeroy Response to RC2**

We appreciate the thoughtful comments and insights provided by Reviewer#2, and below in **bold** include a detailed response to each comment.

I think this is a generally well-written and important paper that attempts to separate the drivers of climate change and vegetation change on hydrology over a historic period. My biggest concern is the observational data used to drive the model, particularly precipitation and snow. I understand that the experiment compares three scenarios (changing climate, changing vegetation, and both) so I would like to see how each of these is impacted by uncertainty in the quality of the precipitation estimates. If the cold season precipitation is most biased, it could be that changes in the length of the cold season cause changes in this bias with time. I think the springtime precipitation trend is suspicious and I'd like to know how uncertainty in those data impacts the robustness of the results. The description of the precip data is useful, but I'd have more confidence if a more thorough comparison of different precip data sources (and nearest other stations) were performed. There are a lot of discontinuities in these datasets, as described. Have the authors considered doing a scenario that separates temperature and precipitation change? In the discussion, I'd like to hear more about impacts of the uncertainty in the precipitation data on the larger results of the study. Same for snow measurements and to a lesser degree, streamflow.

**ANS: The reviewer identifies important aspects of the uncertainty associated with some of the meteorological records used in this study. The measurement of snowfall in this environment is of particular interest to the authors (Pomeroy and Goodison, 1997). Compared to much of northern Canada, uncertainty in precipitation measurement is confined at Inuvik due to the low winter wind speeds and the meticulous data quality control and corrections used in the AHCCD dataset (Mekis and Vincent, 2011). Observations that are not subject to corrections by Mekis and Vincent involve automated weather stations to which well-established wind undercatch corrections were applied to achieve similar corrections. Nevertheless, we agree that the dataset is not perfect and there are discontinuities in the mid-90's when the system changes from manually to automatically observations. Fortunately, snow surveys in sheltered taiga provides a means to evaluate snowfall records, as snow redistribution and sublimation in small clearings in the taiga are minimal and so provide an alternative method to estimating seasonal snowfall**

**in the region (Pomeroy et al., 1997). In the revised version of the manuscript we have provided a more comprehensive discussion about the potential impacts of the uncertainty in precipitation records on the results as per reviewer's suggestion. Unfortunately, there are no nearby stations with similar long-term records to compare the results presented in**

5 **this manuscript. We did not consider a scenario separating temperature and precipitation change, though we note that it would be an interesting exercise, it is out of the scope of the study.**

Minor comments: like the other reviewer, I don't care for the use of the term 'hydrologically resilient' without a technical definition provided. This is too vague. I would also like a little more

10 information on this basin. Is this a well-instrumented research basin? It doesn't really seem like it, based on the description of the single station observations. Why was it chosen? Are there no research basins that fit the description (tundra-taiga boundary with permafrost)? In your introduction, it might be worth mentioning the NASA ABoVE (Arctic-Boreal) campaign, focused on exactly these eco-zones because it has a hydrology component. Finally, while most of

15 the paper is readable, the abstract could use some work. Go for shorter, simpler sentences that really convey what is interesting and exciting about this paper.

**ANS: We agree with the reviewer and have revisited the term "hydrologically resilient". Havikpak Creek has had hydrological studies since 1992 and we now refer more to the detailed description and history of the basin in Krogh et al (2017), and we would like to**

20 **refer to that work instead of repeating those details, with the idea of keeping the manuscript as concise as possible. We have added a paragraph discussing why Havikpak Creek was chosen as the study basin and also include a reference to the NASA ABoVE project as suggested by the reviewer. We have re-written the abstract to make it more readable as per the reviewer's suggestion.**

**Recent Changes to the Hydrological Cycle of an Arctic basin at the Tundra-Taiga Transition**

Sebastian A. Krogh and John W. Pomeroy

Centre for Hydrology, University of Saskatchewan, 121 Research Dr., Saskatoon, SK S7N 1K2, Canada

*Correspondence to:* Sebastian Krogh (seba.krogh@usask.ca)

**Abstract.**

The impact of observed changes in climate and vegetation on the hydrology of Arctic basins is often considered to be most sensitive at the tundra-taiga transition where the region is warmest and sub-arctic vegetation is nearest. This study uses weather and land cover observations and a cold regions hydrological model to investigate historical changes in modelled hydrological processes driving the streamflow response of a small Arctic permafrost underlain basin at the tundra-taiga transition. The physical processes found in this environment and explicit changes in vegetation type and density were simulated and validated against observations of streamflow discharge, snow water equivalent and active layer thickness. Mean air temperature and all-wave irradiance have increased by 3.7°C and 8.4 W m$^{-2}$, respectively, while precipitation has decreased from 369 to 321 mm since 1960. Two modelling scenarios were created to separate the effects of changing climate and vegetation on hydrological processes. Results show that over 1960-2016 most hydrological changes were driven by climate changes, such as decreasing snowfall by 7.8 mm decade$^{-1}$, deepening active layer thickness by 1.8 – 4.2 cm decade$^{-1}$, earlier snowcover depletion and ground thaw initiation dates from 1.5 to 3 and by 1 to 3 days decade$^{-1}$, respectively, and diminishing annual sublimation and soil moisture by 1.3 and 5.9 mm decade$^{-1}$, respectively. Evapotranspiration decreased by 2.5 mm decade$^{-1}$, due to decreasing irradiance and soil moisture. Shrub expansion and densification decreases blowing snow redistribution by 20 to 40 mm and sublimation by 1 to 10 mm. 
[revised manuscript text omitted]

---

## Author Response (AR2)

Dear Editor,

Please find attached our revised version of the manuscript. Changes included in this version correspond to the typos you pointed out and additional typos found during the last revision. Thanks for your help through the discussion process.

5    Best regards,

Sebastian Krogh

[revised manuscript text omitted]